# MicroRNA-218 instructs proper assembly of hippocampal networks

Seth R Taylor[1]*‡, Mariko Kobayashi[2]†, Antonietta Vilella[3]†, Durgesh Tiwari[4,5]†, Norjin Zolboot[6]†, Jessica X Du[6], Kathryn R Spencer[6], Andrea Hartzell[6], Carol Girgiss[1], Yusuf T Abaci[1], Yufeng Shao[6], Claudia De Sanctis[7], Gian Carlo Bellenchi[7,8], Robert B Darnell[2], Christina Gross[4,5], Michele Zoli[3], Darwin K Berg[1], Giordano Lippi[6]*

[1]Division of Biological Sciences, University of California, San Diego, La Jolla, United States; [2]Laboratory of Molecular Neuro-oncology, Howard Hughes Medical Institute, Rockefeller University, New York, United States; [3]Department of Biomedical, Metabolic and Neural Sciences; Center for Neuroscience and Neurotechnology (CfNN), University of Modena and Reggio Emilia, Modena, Italy; [4]Division of Neurology, Department of Pediatrics, Cincinnati Children's Hospital Medical Center, University of Cincinnati College of Medicine, Cincinnati, United States; [5]Department of Pediatrics, University of Cincinnati College of Medicine, Cincinnati, United States; [6]Department of Neuroscience, Scripps Research Institute, La Jolla, United States; [7]Institute of Genetics and Biophysics A Buzzati-Traverso, Naples, Italy; [8]IRCCS Fondazione Santa Lucia, Rome, Italy

*For correspondence:
seth_taylor@byu.edu (SRT);
glippi@scripps.edu (GL)

†These authors contributed equally to this work

Present address: ‡Cell Biology and Physiology, Brigham Young University, Provo, United States

**Competing interest:** The authors declare that no competing interests exist.

**Abstract** The assembly of the mammalian brain is orchestrated by temporally coordinated waves of gene expression. Post-transcriptional regulation by microRNAs (miRNAs) is a key aspect of this program. Indeed, deletion of neuron-enriched miRNAs induces strong developmental phenotypes, and miRNA levels are altered in patients with neurodevelopmental disorders. However, the mechanisms used by miRNAs to instruct brain development remain largely unexplored. Here, we identified miR-218 as a critical regulator of hippocampal assembly. MiR-218 is highly expressed in the hippocampus and enriched in both excitatory principal neurons (PNs) and GABAergic inhibitory interneurons (INs). Early life inhibition of miR-218 results in an adult brain with a predisposition to seizures. Changes in gene expression in the absence of miR-218 suggest that network assembly is impaired. Indeed, we find that miR-218 inhibition results in the disruption of early depolarizing GABAergic signaling, structural defects in dendritic spines, and altered intrinsic membrane excitability. Conditional knockout of *Mir218-2* in INs, but not PNs, is sufficient to recapitulate long-term instability. Finally, de-repressing *Kif21b* and *Syt13*, two miR-218 targets, phenocopies the effects on early synchronous network activity induced by miR-218 inhibition. Taken together, the data suggest that miR-218 orchestrates formative events in PNs and INs to produce stable networks.

## Editor's evaluation

This study describes important work documenting a role of microRNAs in regulating excitability in the developing mouse hippocampus. Strengths of the findings include the identification of several developmentally regulated microRNAs, specific manipulation in principle and inhibitory neurons, and multimodal analysis of developing circuits and gene expression. Solid methods are used to suggest a particular developmental role for interneurons and their regulation by microRNA-218.

## Introduction

Network activity levels in the adult brain are determined by the precise interplay between excitatory pyramidal neurons (PNs) and GABAergic inhibitory interneurons (INs; *Isaacson and Scanziani, 2011*). A properly balanced network is achieved through a series of developmental events in early post-natal life (*Blankenship and Feller, 2010*). During neural circuit assembly, neurons first rapidly grow dendrites and axons to form a coarse network, followed by a synaptic refinement period shaped by sensory experience (*Blankenship and Feller, 2010*). At this early stage, the interplay between PNs and INs is crucial for cell survival, growth, and synapse formation. For example, depolarizing GABA released from INs generates synchronous activity that drives gene expression programs to instruct PN growth and network assembly (*Ben-Ari, 2002*; *Bonifazi et al., 2009*; *Cancedda et al., 2007*; *Deidda et al., 2015*). Conversely, excitatory PN afferents sustain the survival of INs and mediate their integration into the circuit (*Wong et al., 2018*; *Xue et al., 2014*). Unsurprisingly, errors at this stage have catastrophic long-term consequences for network function. Derailment from the healthy developmental trajectory results in aberrant levels of neuronal activity in the adult brain, a feature that can disrupt the precise computations underlying cognition and behavior. In fact, aberrant network activity is a core symptom of neuropsychiatric disorders, including epilepsy, autism spectrum disorder, and schizophrenia (*Rubenstein and Merzenich, 2003*; *Zoghbi and Bear, 2012*).

At the molecular level, circuit assembly is orchestrated by a multitude of genes whose expression is tightly regulated in time (developmental stages) and space (cell types). Studies spanning the past few decades have identified sophisticated transcriptional regulation of these genes (*Silbereis et al., 2016*; *Telley et al., 2016*). However, the post-transcriptional regulation of gene expression is less well understood. This regulation is mediated by a class of non-coding RNAs, miRNAs. MiRNAs either degrade mRNA targets by reducing their stability or repress their translation, providing a layer of control of gene expression that supersedes transcriptional activation (*Bartel, 2018*; *Bartel, 2004*). This post-transcriptional regulation is necessary for neuronal development (*Ebert and Sharp, 2012*; *McNeill and Van Vactor, 2012*): conditional knockout (cKO) of *Dicer*, an essential component of the miRNA biogenesis machinery, induces a marked decrease in neuronal survival and multiple defects in both cortical lamination and the formation of proper synaptic connections (*De Pietri Tonelli et al., 2008*; *Kawase-Koga et al., 2009*; *Tuncdemir et al., 2015*; *Zolboot et al., 2023*). We and others have shown that deletion of a single highly enriched neuronal miRNA is sufficient to induce strong developmental phenotypes, including disruption of circuit assembly (*Dulcis et al., 2017*; *Lippi et al., 2016*; *Lippi et al., 2011*; *Schratt et al., 2006*; *Siegel et al., 2009*; *Siegert et al., 2015*; *Tan et al., 2013*; *Zolboot et al., 2023*). Together, these studies suggest that miRNAs tightly control key facets of neuronal development. Unsurprisingly, patients suffering from neurodevelopmental disorders have altered levels of miRNAs (*Bian and Sun, 2011*; *O'Connor et al., 2016*; *Wu et al., 2016*), but if and how miRNAs contribute to disease is unknown. Both miRNAs and target mRNAs can be enriched in specific cell types, generating cell type-specific miRNA-target networks (*He et al., 2012*; *Nowakowski et al., 2018*). In cortical structures most of the work has focused on PNs. However, recent research has demonstrated crucial roles for miRNAs in INs, from early development (survival, specification, and lamination of medial ganglionic eminence-derived INs) to cognitive functions and memory formation (*Daswani et al., 2022*; *Qiu et al., 2020*; *Tuncdemir et al., 2015*).

Previous work has demonstrated that miR-218 plays a crucial role in cell fate specification and survival of embryonic motor neurons (*Amin et al., 2021*; *Amin et al., 2015*; *Reichenstein et al., 2019*; *Thiebes et al., 2015*) as well as in regulating cognitive functions (*Lu et al., 2021*; *Torres-Berrío et al., 2021*). Here, we show that miR-218 is essential for multiple aspects of hippocampal circuit assembly. Blockade of miR-218 in early postnatal life results in networks with excessive neuronal activity in the adult, a symptom common to many neurodevelopmental disorders. Our data suggest that miR-218 regulates several key processes during early postnatal life that are instrumental to produce a balanced network. These include early depolarizing GABAergic signaling, structural growth of spines, and the regulation of intrinsic membrane excitability. Finally, we show that cKO of *Mir218-2* in INs, but not PNs, recapitulates long-term effects on circuit stability, suggesting an important role in this cell type.

# Results

## MiR-127, miR-218, and miR-136 are highly expressed in the hippocampus and developmentally upregulated

A wealth of literature indicates that miRNAs regulate key neurodevelopmental processes that build well-balanced, properly functioning brains (*Zampa et al., 2019*; *Zolboot et al., 2021*). Here, we aim to identify miRNAs that are necessary for hippocampal network development in early postnatal life and for stability in the adult. We chose the hippocampus as a model system because its circuitry and function are well known and because errors in hippocampal development lead to neurodevelopmental disorders (*Li et al., 2019*), such as temporal lobe epilepsy (*Bender et al., 2004*). Small RNA-Sequencing (RNA-Seq) of the dorsal hippocampus at P12 revealed abundant expression of many miRNAs (*Lippi et al., 2016*). We selected 10 highly expressed candidate miRNAs and used qRT-PCR to determine their expression levels before (embryonic day 16, E16 and postnatal day 3, P3), during (P12), and after (P40-48) hippocampal circuit assembly. Three candidates, miR-127, miR-218, and miR-136, increase substantially in expression during early postnatal development, suggesting that they function in this developmental window (*Figure 1A–B*). MiR-218 levels increased nearly threefold from E16 to P12 and remained elevated into adulthood. MiR-127 and miR-136 increased robustly between E16 and P12 (three- and sixfold, respectively), but declined in the adult.

## miR-218 inhibition produces excessive immature network activity

To test if miR-127, miR-218, and miR-136 are important for hippocampal network development, we designed antagomiRs, modified antisense oligonucleotides that induce the knockdown of a specific miRNA (a.127, a.218, a.136, *Figure 1C–D*). AntagomiRs are fluorescent, have long half-lives, and readily enter all cells in the injected area. We have pioneered their use in the developing brain, and extensively tested their efficacy and specificity (*Dulcis et al., 2017*; *Lippi et al., 2016*). During early postnatal life, large, synchronous waves of depolarizing activity are key drivers of circuit assembly (*Ben-Ari, 2002*; *Ben-Ari et al., 1989*; *Cossart and Khazipov, 2022*; *Leinekugel et al., 2002*). We have shown that altered levels of these synchronous events (SEs) are a reliable indicator of derailment from the proper developmental trajectory that results in circuit miswiring and aberrant activity later in life (*Lippi et al., 2016*). We asked whether early postnatal inhibition (starting at P2) of miR-127, miR-218, or miR-136 alters SEs compared to a non-targeting antagomiR control (a.Ctrl). We performed calcium imaging of acute hippocampal slices at P8 using the cell-permeable calcium indicator dye Fluo-4AM. We imaged cells in the CA3 subregion, which is comprised of 80–90% PNs and 10–20% INs. The CA3 subregion was chosen because of its role in generating hippocampal synchronies that in pathological conditions can lead to seizures (*Cherubini and Miles, 2015*). Inhibition of miR-218 and, but not of miR-127 or miR-136, resulted in increased frequency of SEs, a sign of excessive immature network activity (*Figure 1E–F*). Inhibition of miR-218, miR-127, or miR-136 did not significantly change total activity (*Figure 1G*). The percentage of cells firing per SE and of active cells (*Figure 1H–I*) were not affected. These results indicate that miR-218 activity is involved in establishing early neuronal activity patterns.

## Transient miR-218 inhibition in early life produces excessive network activity and selective cognitive problems in the young adult

MiR-218 was previously found to be down-regulated in the hippocampus of patients with mesial temporal lobe epilepsy and hippocampal sclerosis (*Kaalund et al., 2014*) and to regulate neuronal excitability in the spinal cord and in cultured neurons (*Amin et al., 2015*; *Lu et al., 2021*; *Reichenstein et al., 2019*). If miR-218 is necessary for circuit assembly in early postnatal life, inhibition should cause long-term deficits in the adult hippocampus. We co-injected a.218 or a.Ctrl with an adeno-associated virus driving the calcium indicator GCaMP6f into the dorsal hippocampus at P2 and allowed mice to live until young adulthood (*Figure 2A*). We monitored hippocampal activity by calcium imaging of the CA3 pyramidal layer in acute slices from mice at P42 to P46. Inhibition of miR-218 did not result in significant changes in spontaneous activity in CA3 PNs (*Figure 2B–C*).

Changes in miR-218 levels have been described in patients with temporal lobe epilepsy (*Kaalund et al., 2014*). Hence, we asked whether early blockade of miR-218 predisposes adult networks to seizure activity. To test this, we injected the dorsal hippocampus at P2 with a.218 or a.Ctrl and implanted

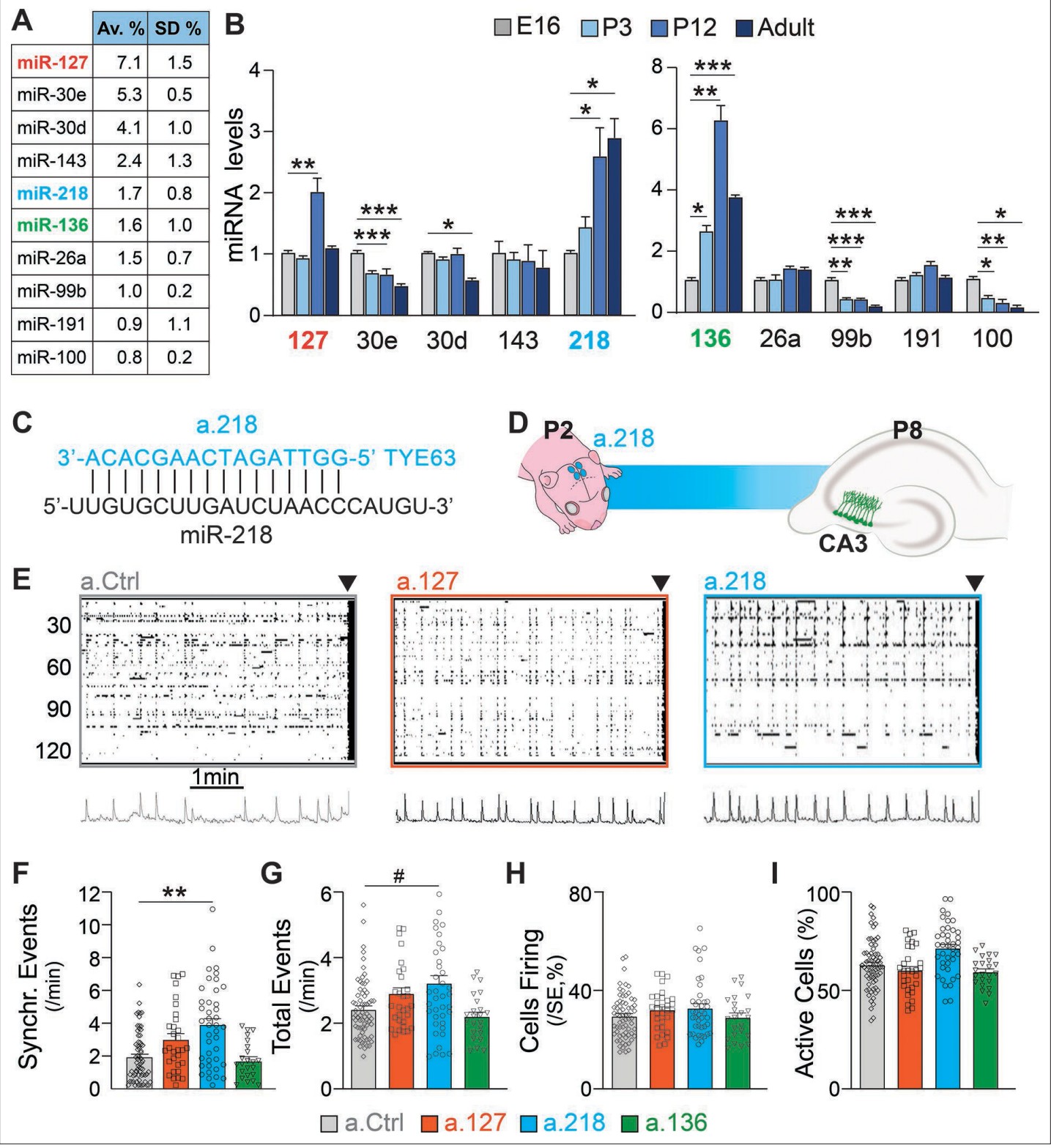

**Figure 1.** Inhibition of the developmentally regulated miRNA miR-218 alters spontaneous neuronal activity in early postnatal life. (**A**) Abundance of a subset of highly expressed miRNAs in the developing hippocampus (data averaged for 6 C57BL/6J P12 animals) expressed as percentage of total sequenced reads obtained from small RNA deep sequencing (Illumina). Data come from *Lippi et al., 2016*. N=6 mice. (**B**) Developmental regulation of 10 abundant miRNAs measured by miRNA qPCR. Bar graphs represent the abundance of each miRNA at P3, P12, and adult (>P40), normalized first for housekeeping gene *Snord45b* (available from Thermofisher under name: snoRNA412) and then for the amount present at E16. MiR-127, miR-218,

*Figure 1 continued on next page*

*Figure 1 continued*

and miR-136 increased in expression during early postnatal development. Note: well-studied miRNAs (i.e. miR-124) were not tested here. N=3–5 / time point. Statistics were calculated from a full mixed effects model with miRNA and age as fixed effects and animal as a random effect, followed by post-hoc tests with FDR multiple testing correction. (**C**) Illustration showing the complementarity of a-218 with miR-218 (vertical bars indicating the complementary bases) which is essential for the inhibition. (**D**) Schematic of the time course of miR-218 inhibition, starting at P2, and calcium imaging performed at P8 in the CA3 hippocampal region. (**E**) Raster plots of neuronal activity in movies of animals injected with a.Ctrl (left), a.127 (center), and a.218 (right). The principal component analyses of population activity are shown below to identify the SEs. (**F–I**) MiR-218 inhibition increased the number of SEs (**F**, p=0.0019 for miR-218). MiR-218 inhibition increased total events (**G**, p=0.0625), but did not change the percentage of cells in SEs or that are active (**H–I**). MiR-136 or miR-127 inhibition had no effect on spontaneous activity. Bar graphs: mean ± SEM; Outliers detected with ROUT in GraphPad Prism (Q=1%) and removed. Statistics were calculated from a linear mixed model accounting for animal as a random effect, with Dunnett's multiple comparisons test. #p<0.1, *p<0.05, **p<0.01, ***p<0.001, N for a.Ctrl=69 movies/19 animals; a.136=23 movies/5 animals; a.127=31 movies/7 animals; a.218=41 movies/8 animals.

single-lead cortical surface electrodes and wireless transmitters at around P30 to enable 24/7 electro-encephalographic (EEG) recordings. Between P42 and P46, we injected mice intraperitoneally with kainic acid (KA, 15 mg/kg) and monitored network activity. Early, transient inhibition of miR-218 significantly decreased the time to seizure onset following KA injection (*Figure 2D–E*). EEG power analysis of different frequency bands during the KA-induced seizures revealed that miR-218 inhibition also increased seizure severity, as evidenced by significant increases in the alpha, beta, gamma, and sigma bands of the cumulative EEG waveforms in a.218 compared to a.Ctrl mice (*Figure 2D and F*). EEG waveforms were not altered before the KA challenge (*Figure 2—figure supplement 1A*). Increases in EEG waveforms gamma, theta and beta power have been observed during the onset of convulsive seizures (*Maheshwari et al., 2016*; *Marrosu et al., 2006*; *Phelan et al., 2015*; *Tiwari et al., 2020*; *Tse et al., 2014*) and in human epilepsy (*Perry et al., 2014*). Combined with our calcium imaging results, these data indicate that early inhibition of miR-218 results in adult hippocampal networks with excessive activity that are susceptible to seizures after a second insult (KA). Therefore, miR-218 is necessary for the development of networks with stable levels of neuronal activity.

To assess if the a.218-induced alterations in neuronal activity observed in the young adult had functional consequences, we performed a battery of behavioral tests. To test spatial learning, we used the Morris water maze (*Daini et al., 2021*). Inhibition of miR-218 did not alter the latency to find the platform (*Figure 3A*) but did result in a significant shift in the strategies used to reach the platform. While in the first 9 days of test (D1-D9, *Figure 3B*), roughly half of a.218 and a.Ctrl mice used a spatially targeted strategy to reach the goal, at D11 and during the final probe test (D12) ~90% of a.Ctrl mice used a spatial strategy to reach platform location whereas half of a.218 mice continued to use non-spatial / random (non-cognitive) swim strategies to reach the platform area (*Figure 3B*). No differences were observed in the learning curve or the ability to discriminate between similar contexts. To assess compulsive/perseverative behaviors, we performed the marble burying test (*Daini et al., 2021*). a.218 mice showed no significant changes in the number of buried marbles (*Figure 3C*) but a reduced latency to the first burying (*Figure 3D*) compared to a.Ctrl mice. Inhibition of miR-218 did not change spontaneous exploration (open field, *Figure 3—figure supplement 1A*), anxiety (elevated plus maze, *Figure 3—figure supplement 1B*), working memory (Y maze, *Figure 3—figure supplement 1C*), sociability (three-chamber sociability test, *Figure 3—figure supplement 1D*), or stereotypic behaviors (rearing, grooming and digging) (*Figure 3—figure supplement 1E*). These data suggest that transient inhibition of miR-218 during early postnatal life causes selective cognitive deficits and repetitive behaviors.

## MiR-218 expression pattern and timing of effective inhibition with a.218

To understand which developmental events are disrupted in a.218 injected animals, we characterized miR-218 expression and identified the temporal window of functional miR-218 inhibition with a.218. In situ hybridization revealed abundant expression in the CA1 pyramidal layer and dentate gyrus with lower levels in the CA3 pyramidal layer (*Figure 4A*). Fluorescent in situ hybridization combined with immunohistochemical staining revealed abundant expression in neurons with less abundant expression in glia (*Figure 4B*). Further, analysis of public miRNA expression and Ago2 IP databases (*He et al., 2012*) revealed that miR-218 is abundant in both CaMKII +PNs and PV +INs, while expression

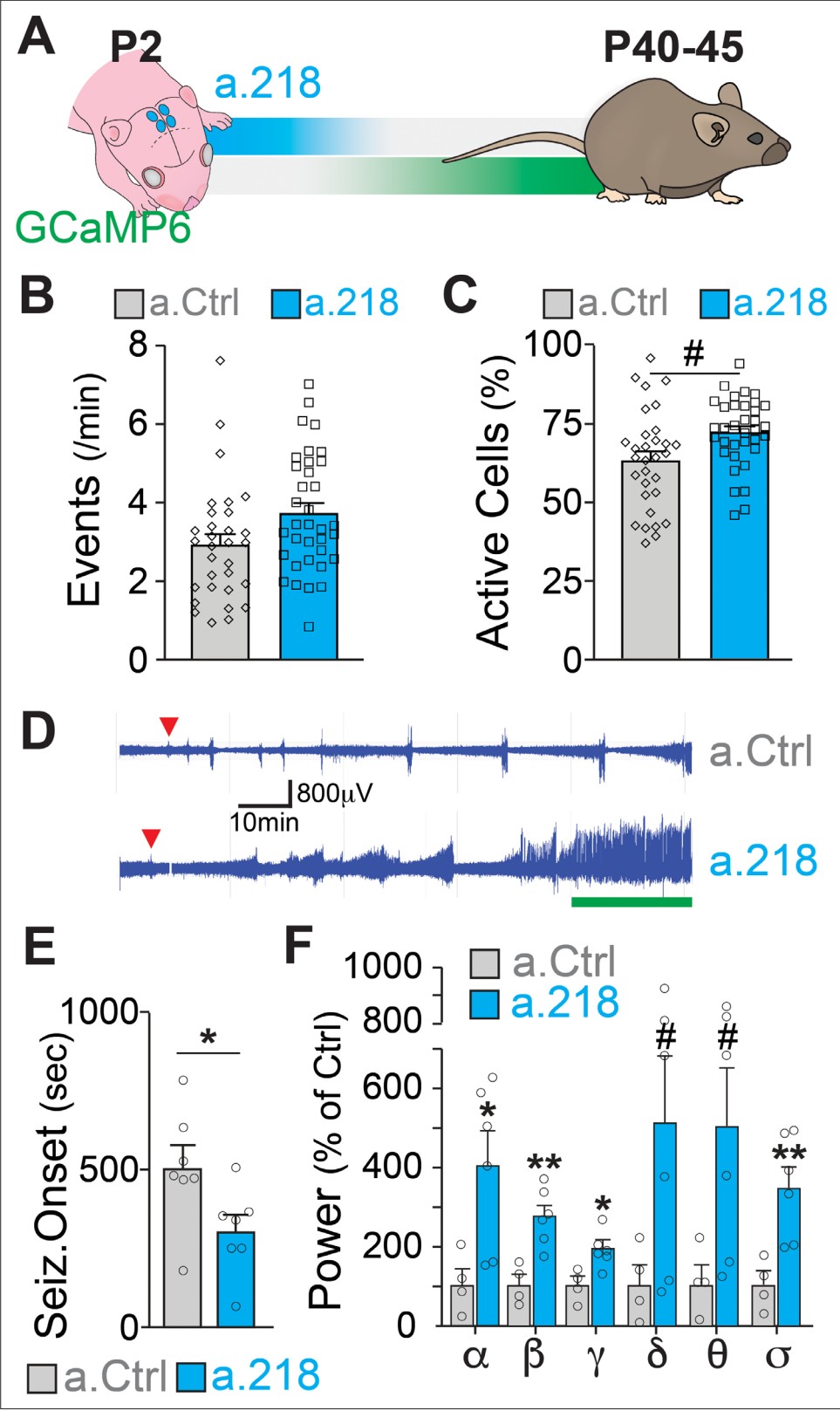

**Figure 2.** MiR-218 inhibition in early postnatal life causes long-lasting deficits in network activity. (**A**) Schematic of the timeline of miR-218 inhibition and of GCaMP6f expression. (**B**) Quantification of calcium activity showed no changes in the number of events (p-value = 0.1976) and (**C**) an increase in the percentage of active cells (p-value = 0.05562) in slices from P40-45 animals injected on P2. For B and C, linear mixed model accounting for animal as

*Figure 2 continued on next page*

*Figure 2 continued*

a random effect. N for a.Ctrl=31 movies from 5 animals, N for a.218=35 movies from 5 animals. (**D–F**) KA-induced seizures at ~P40 start earlier in a.218-treated mice. (**D**) EEG recordings immediately after KA. a.218 mice show stronger responses. The green line denotes continuous status epilepticus, a sign of a pathological network activity. Red triangle: first seizure (confirmed by behavior). a.218 mice show faster seizure onset, N=7 mice/group, unpaired t-test (**E**) and increased power at many frequencies (**F**), N=4–6mice/ group, unpaired t-tests. Bar graphs: mean ± SEM; #p<0.1, *p<0.05, **p<0.01. α: alpha, β: beta, γ: gamma, δ: delta, θ : theta, σ: sigma.

The online version of this article includes the following figure supplement(s) for figure 2:

**Figure supplement 1.** Early life inhibition of miR-218 has no effect on baseline power before the KA challenge. N=4–6mice/ group. Bar graphs: mean ± SEM; α: alpha, β: beta, γ: gamma, δ: delta, θ : theta, σ: sigma.

levels in astrocytes, oligodendrocytes, and microglia are noticeably lower (*Figure 4—figure supplement 1A*). MiR-218 is therefore enriched in both PNs and INs, but not in glia.

Mature miR-218 is encoded by two genes, *Mir218-1* and *Mir218-2*, located on different chromosomes in the introns of axon guidance genes *Slit2* and *Slit3* (*Tie et al., 2010*; *Figure 4C*). The

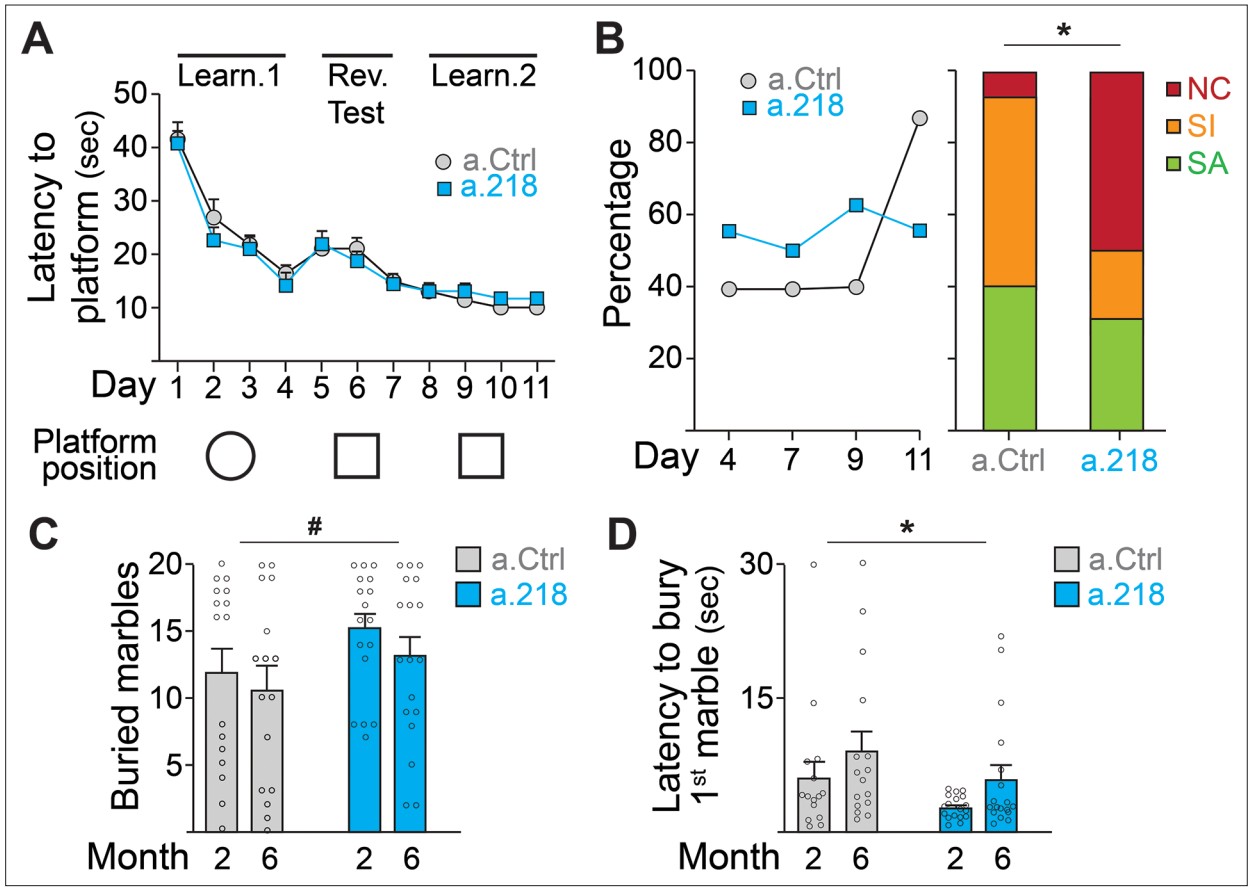

**Figure 3.** MiR-218 inhibition causes selective behavioral changes. (**A–B**) Morris water maze. (**A**) The learning curve did not demonstrate any statistically significant differences between experimental groups (repeated-measures ANOVA, Time F (10, 290) p=0.0001, Treatment F (1, 29) p=0.7167, Time x Treatment F (10, 290) p=0.7843). The ability to discriminate between similar contexts (reversal test, days 5 and 6) appeared preserved in a.218 mice (repeated-measures ANOVA, Time F (1, 29) p=0.7595, Treatment F (1, 29) p=0.2183, Time x Treatment F (1, 29) p=0.3450). Similarly, the latency for the first entry on platform location in the probe test did not differ between the two groups (p=0.3862). (**B**) Search strategy analysis showed an increase in spatially targeted strategies during learning phase and probe test for a.Ctrl compared to a.218 mice (Fisher's test, p=0.02). NC: non cognitive; SI: spatial inaccurate; SA: spatial accurate. (**C–D**) Marble burying Test. a.218 mice showed an increase in the number of buried marbles (**C**) (2 months vs 6 months; repeated measures ANOVA, F (1, 31)=4.074, p=0.052) and a reduced latency to bury the first marble (**D**) (2 months vs 6 months; repeated measures ANOVA, F (1, 31)=6.257, p=0.018). Bar graphs: mean ± SEM; #p<0.1, *p<0.05. N=16–18 mice/group.

The online version of this article includes the following figure supplement(s) for figure 3:

**Figure supplement 1.** Early life inhibition of miR-218 has no effect on several behavioral measures.

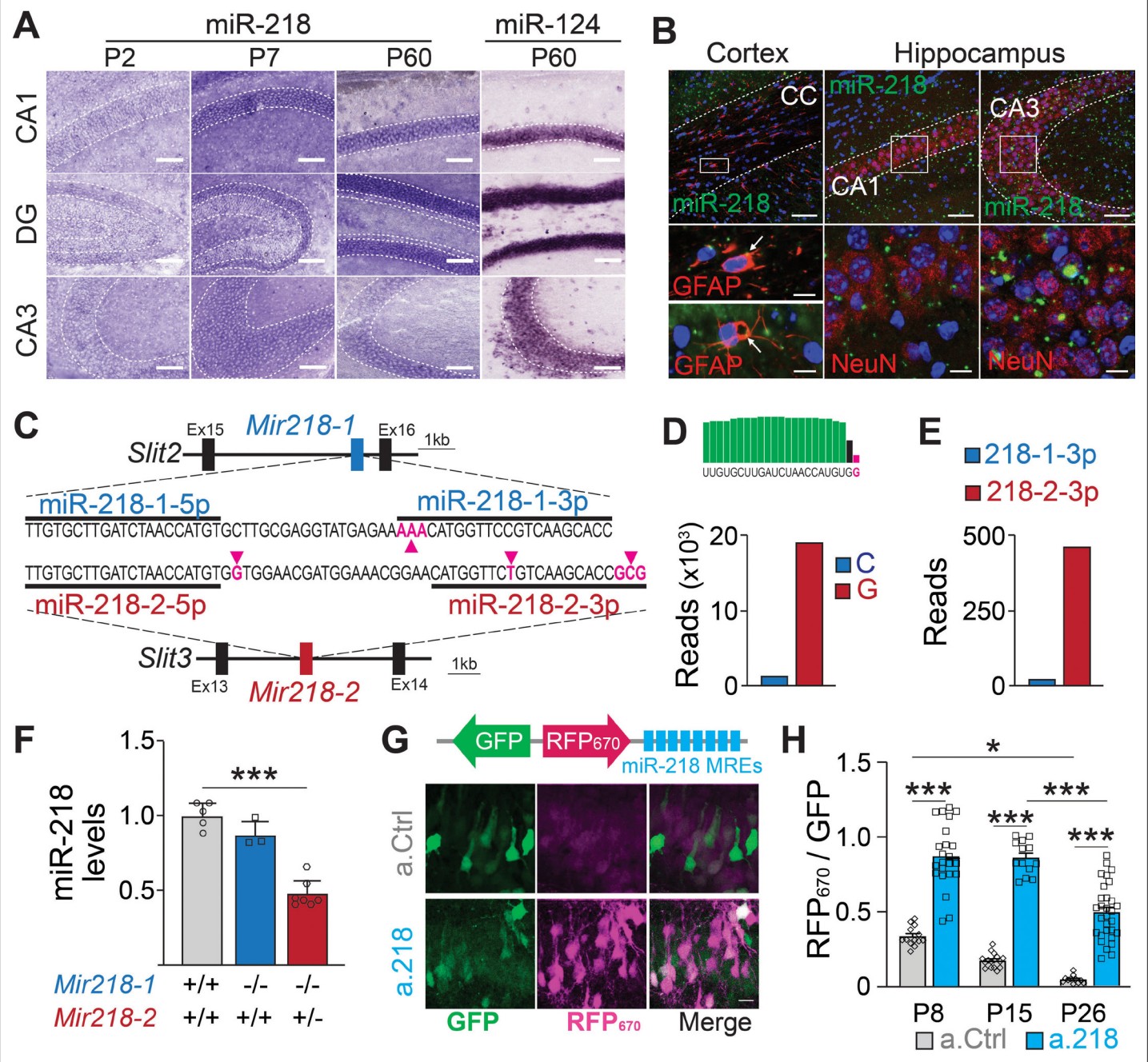

**Figure 4.** Spatial and temporal distribution of miR-218 expression and inhibition. (**A–B**) MiR-218 and miR-124 expression in developing mouse brain. (**A**) Optical microscopy images of fresh frozen brain slices labelled with miR-218 and miR-124 LNA ISH probes represented by dark staining due to NBT/BCIP precipitation. MiR-218 is detected in the CA1, DG, and CA3 regions of the hippocampus. MiR-218 expression changes during hippocampal maturation from a low level at P2 that increases at P7 and then remains stable in the adult (P60). MiR-124, a highly abundant neuronal miRNA, is used at P60 as positive control. Scale bar = 100 μm. (**B**) Confocal microscopy images of fresh frozen brain slices labelled with miR-218 LNA probes (green), DAPI (blue) and anti-GFAP or anti-NeuN antibodies (red). Top: representative panels of miR-218 expression in cortex, corpus callosum (CC) and hippocampus (CA1, CA3). Scale bar = 50 μm. Bottom left: high magnification images showing single cells from the corpus callosum. Green and red colocalization is marked by yellow color (arrows). Bottom right: high magnification images showing neuronal cells from CA1 and CA3. Green spots are localized inside neuronal cells labelled in red. Scale bar = 10 μm. (**C**) Schematic of *Mir218-1* gene located in *Slit2* (intron 15) and *Mir218-2* gene located in *Slit3* (intron 13). The prevalent mature miR-218 sequence (5 p) is identical between the two genes. However, the second base downstream the 3' end of the 5 p sequence is G in *Mir218-2* (purple arrow) and C in *Mir218-1*. The less abundant miR-218–3 p sequences differ in multiple bases between the genes (purple arrows). (**D**) Small RNA-Seq data were used to detect C or G at the second base downstream the 3' end of the 5 p sequence. G (*Mir218-2*) is much more frequent. (**E**) Similarly, miR-218-2-3p is more abundant than miR-218-1-3p. (**F**) MiR-218 levels in P15 *Mir218-1* KO animals are

*Figure 4 continued on next page*

*Figure 4 continued*

not significantly different from WT controls. In contrast, loss of one *Mir218-2* allele causes a 50% reduction in miR-218 levels. N=3–7 animals/genotype, adjusted p=0.002, Kruskal-Wallis test with Dunn's multiple comparisons test. (**G**) Lentiviral miRNA sensors demonstrate that miR-218 is abundant in vivo and can be blocked by a.218. Top, scheme of the lentiviral construct encoding the miR-218 sensor (8 x perfectly complementary miR-218 MREs cloned downstream of a far-red fluorescent protein, $RFP_{670}$). Bottom, CA1 region in mice injected with the sensor and either non-fluorescent a.Ctrl (top) or a.218 (Bottom). a.218 rescues $RFP_{670}$ fluorescence. Scale bar = 10 μm. (**H**) Quantification of $RFP_{670}$ normalized for GFP. All values are expressed relative to a control sensor lacking miR-218 MREs in which $RFP_{670}$ fluorescence is not repressed by miR-218 ($RFP_{670}$ /GFP = 1). 8 x miR-218 MREs strongly reduce $RFP_{670}$ fluorescence in the presence of a.Ctrl. $RFP_{670}$ fluorescence declines with age, reflecting the developmental increase in miR-218 expression. At P8 and P15, a.218 rescues $RFP_{670}$ fluorescence to levels similar to the control sensor, indicating effective inhibition of miR-218. The inhibition is strongly decreased by P26. At P8, N=14 sections/3 animals (a.Ctrl) and 23 sections/3 animals (a.218). At P15, N=16 sections/3 animals (a.Ctrl) and 15 sections/4 animals (a.218). At P26, N=11 sections/4 animals (a.Ctrl) and 31 sections/6 animals (a.218). Bar graphs: mean ± SEM; linear mixed model with age and antagomir as fixed effects accounting for animal as a random effect with Tukey adjustment for multiple comparisons. *p<0.05, ***p<0.001.

The online version of this article includes the following figure supplement(s) for figure 4:

**Figure supplement 1.** MiR-218 is enriched in CaMKII +PNs and PV +INs.

predominant mature miRNA sequence (miR-218–5 p) from both genes is identical, but surrounding nucleotides and the 3 p species from the precursor molecules differ between the genes (*Amin et al., 2015*). Small RNA-Seq reads containing these surrounding nucleotides were 18 times more abundant from *Mir218-2* than from *Mir218-1* (*Figure 4D*). RT-PCR from hippocampi of *Mir218* constitutive mutant mice confirmed these findings. *Mir218-1* KO mice (*Mir218-1$^{-/-}$*; *Mir218-2$^{+/+}$*) had similar levels of mature miR-218 as wild-type (WT) mice (*Mir218-1$^{+/+}$*; *Mir218-2$^{+/+}$*). Because complete double KO of *Mir218* is lethal (*Amin et al., 2015*), we used *Mir218-2* hemizygous mice (*Mir218-1$^{-/-}$*; *Mir218-2$^{+/-}$*) and observed a 50% reduction in mature miR-218 compared to WT littermates (*Figure 4F*). These findings strongly suggest that mature miR-218 in the developing hippocampus is produced almost entirely from the *Mir218-2* gene locus.

We validated the efficacy and timing of miR-218 knockdown with a.218 with a miRNA sensor (*Han et al., 2016*; *Lippi et al., 2016*). We previously characterized the spread of antagomiRs in the developing hippocampus (*Lippi et al., 2016*). We utilized a bicistronic lentiviral construct driving a miR-218-insensitive GFP and the far-red fluorescent protein $iRFP_{670}$ containing 8 x miR-218 miRNA recognition elements (MREs) in the 3' UTR. MiR-218 suppresses $iRFP_{670}$ expression, hence the ratio of $iRFP_{670}$ to GFP serves as a readout of miR-218 activity (*Figure 4G*). As a control, we used the same sensor lacking the miR-218 MREs. We co-injected sensor (control or miR-218) and either a.218 or a.Ctrl into the dorsal hippocampus of P2 pups and monitored miR-218 activity at P8, P15, and P26. Consistent with increasing levels of miR-218 expression during development, the miR-218 sensor with a.Ctrl showed decreasing $iRFP_{670}$/GFP ratios from P8 to P26. Injection of a.218 at P2 restored the $iRFP_{670}$/GFP ratio in the miR-218 sensor to a similar level as the control sensor at both P8 and P15 (*Figure 4G–H*). A weaker partial inhibition of miR-218 activity was still present at P26. These results indicate that a.218 injection at P2 strongly inhibits miR-218 activity in vivo for at least two weeks.

## miR-218 target identification suggests possible mechanisms of excessive neuronal activity

The first 2 weeks of postnatal life are a time of tumultuous changes in the developing mouse brain. Neurons must complete their morphological differentiation and find the appropriate synaptic partners. Defects in many different formative events could result in excessive network activity. We reasoned that identifying the transcriptional consequences of miR-218 inhibition would help delineate the impaired developmental processes leading to long-term deficits. De-repression of miR-218 targets likely induces a cascade of transcriptional changes (both up-and down-regulated genes) that could be responsible for the phenotypes observed. To have a broad picture of the molecular changes induced by a.218, we performed RNA-Seq from the dorsal hippocampus of P8 mice treated with a.218 compared to a.Ctrl-treated littermates. We compiled a list of predicted miR-218 targets from the TargetScan prediction database and a published CLEAR-CLIP dataset from P13 mouse cortex (*Moore et al., 2015*). This set of putative miR-218 targets showed a modest, but significant increase in expression in a.218 hippocampus compared to a.Ctrl (*Figure 5A*). This is consistent with miRNAs primarily affecting RNA stability of their targets. However, the relatively small size of the increase might also reflect miR-218 targets that are repressed translationally and therefore are not captured by RNA-Seq.

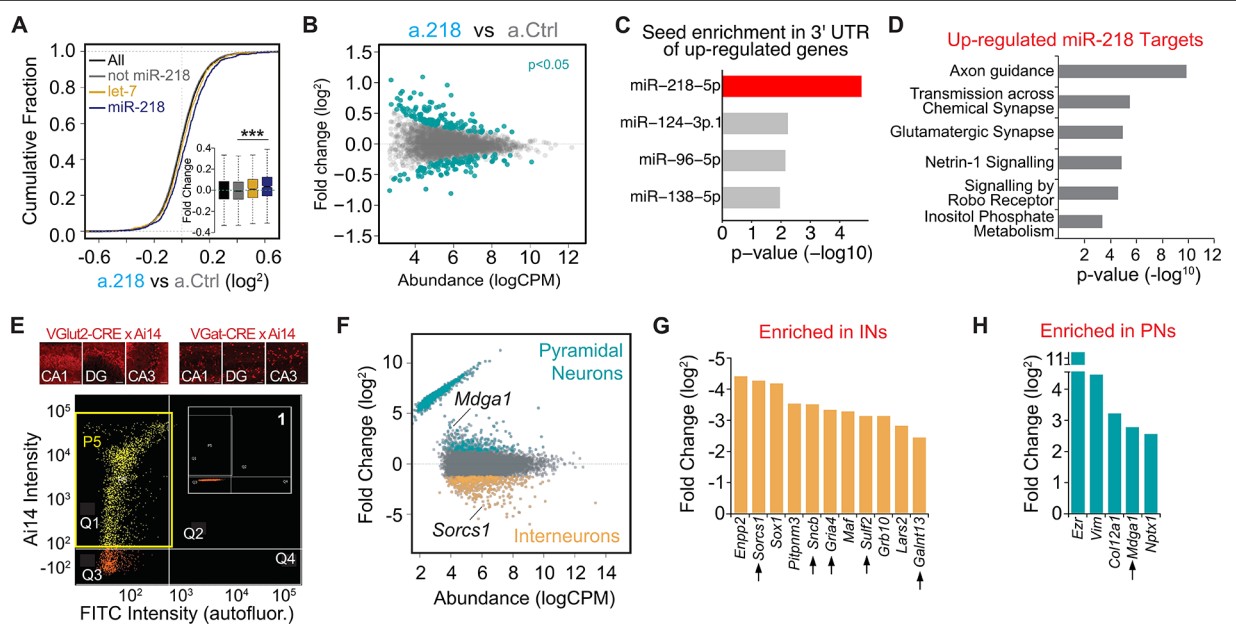

**Figure 5.** Mapping miR-218 targets and transcriptional changes upon miR-218 inhibition with RNA-Seq from dorsal hippocampus at P8. (**A**) Cumulative fraction of all genes (black line), non miR-218 targets (gray line), let-7 targets (yellow), miR-218 targets purple. Fold change of this subset of genes in the insert (bottom right). ***p<9.7e-12 (KS-test). (**B**) MA plot (Abundance vs Fold Change) of transcriptional changes in a.218 mice compared to a.Ctrl littermates. Green dots represent upregulated (top, p<0.05) or downregulated genes (bottom, p<0.05). N=3 mice/group. (**C**) MiRNA seed enrichment analysis of up-regulated genes (p<0.05). The MRE for miR-218 is significantly enriched, suggesting that the list of upregulated genes contains bona fide miR-218 targets. (**D**) GO term analysis of all upregulated predicted miR-218 target genes (391 genes). (**E**) FACS of VGluT2-CRE/Ai14 or VGAT-CRE/Ai14 dorsal hippocampi. Top: representative images of Ai14 positive PNs (top row) or INs in CA1, DG, and CA3 at P8. Bottom: two-dimensional scatterplot illustrating FACS isolation of PNs. Ai14 fluorescence intensity is displayed on the Y axis and FITC (autofluorescence) is on the X axis. Sorted VGluT2-CRE/Ai14 PNs are shown as yellow dots. The P5 yellow rectangle (in quarter Q1) delimits the area chosen for sorting. Insert (1) shows the same gating for CRE negative neurons, where the P5 gate is empty. Scale bar = 50 µm. (**F**) MA plot of transcriptional differences between PNs (VGluT2-CRE/Ai14) and INs (VGAT-CRE/Ai14) showing genes significantly up-regulated in PNs and INs. (**G–H**) Genes upregulated in a.218 vs a.Ctrl (p<0.05) and significantly enriched (FDR <0.05) in either INs (**G**) or PNs (**H**). Black arrows denote predicted miR-218 targets.

The online version of this article includes the following figure supplement(s) for figure 5:

**Figure supplement 1.** Validation of a.218 and identification of miR-218 targets.

Further, we identified subsets of genes that were either up-regulated (138 genes, p-value <0.05) or down-regulated (102 genes, p-value <0.05) in a.218 compared to a.Ctrl (*Figure 5B*, *Supplementary file 1A*). The up-regulated genes showed significant enrichment of the miR-218 seed (*Figure 5C*), as well as additional miRNA seeds, suggesting convergence of repression by multiple miRNAs.

To identify biological processes regulated by miR-218, we performed gene ontology (GO) analysis on all predicted miR-218 targets with log fold change (logFC) >0 (391 genes) in a.218 vs a.Ctrl. This revealed enrichment of genes involved in several biological processes relevant to neural circuit development and activity, including axon guidance, transmission across chemical synapse, and glutamatergic synapse (*Figure 5D*). We repeated the same analysis on a subset of up-regulated genes (logFC >0.1, p-value <0.2), irrespective of being miR-218 targets. Enriched biological processes, among others, were dendritic transport, postsynaptic density assembly, potassium ion homeostasis, and axonogenesis (*Figure 5—figure supplement 1C*). In parallel, we performed RNA-Seq analysis from the dorsal hippocampus of *Mir218-1*$^{-/-}$; *Mir218-2*$^{+/-}$ mice at P15, animals that have 50% reduction in mature miR-218 compared to WT littermates (*Figure 4F*). These animals showed more pronounced transcriptional changes (*Figure 5—figure supplement 1D*, *Supplementary file 1B*), consistent with a constitutive hemizygosity for *Mir218*.

Although up-regulated genes were also enriched for the miR-218 seed, among other miRNA seeds, (*Figure 5—figure supplement 1E*), the list of predicted miR-218 targets did not show an increase in expression (*Figure 5—figure supplement 1F*), underscoring a cascade of complex transcriptional changes following the loss of 50% of miR-218 in all cells for the entire life of the animals. However, we

identified a highly significant overlap in the list of predicted miR-218 targets with logFC >0 between the a.218 and *Mir218-2*[+/-] experiments (*Figure 5—figure supplement 1G*).

Because miR-218 is enriched in both PNs and INs, we compared transcriptomes of INs and PNs from P8 hippocampi to identify targets enriched in each cell type. We performed fluorescence activated cell sorting (FACS) of VGAT-Cre and VGluT2-Cre mice crossed with Ai14, a conditional fluorescent reporter, to isolate INs and PNs separately at P8 (*Figure 5E*, *Supplementary file 1C-D*). RNA-Seq of the sorted cells revealed mRNAs highly enriched in each cell type (*Figure 5F*). Cross-referencing this experiment with the a.218 vs a.Ctrl RNA-Seq experiment performed above, we found that the majority of genes affected by a.218 are expressed in both cell types, but a few are enriched in INs or PNs, suggesting that regulation by miR-218 is biased to that cellular context (*Figure 5G and H*). In addition, we compared the up-regulated genes (*P*<0.05, 138 genes) from a.218 vs a.Ctrl RNA-Seq with a published single-cell RNA-Seq dataset from early postnatal (P2 and P11) mouse brain (*Rosenberg et al., 2018*). The up-regulated genes showed significant enrichment in several developing interneuron populations and in hippocampal CA3 pyramidal neurons (*Figure 5—figure supplement 1H*). Circuit formation arises from the coordinated interaction of activity-dependent and intrinsic formative events. Our RNA-Seq data revealed that inhibition of miR-218 de-represses genes that regulates multiple aspects of this process. We therefore hypothesized that the increased network activity measured in the adult could result from defective inhibition, excessive excitatory drive, or altered intrinsic excitability.

## MiR-218 inhibition affects proper synaptic architecture

Inhibition of miR-218 changes the expression levels of many genes involved in synaptogenesis (*Figure 5D*). To understand if synaptic rearrangements underlie a.218-induced phenotypes, we examined the role of miR-218 activity on synapse formation within the hippocampus using immunohistochemistry and electrophysiology. In all experiments, a.Ctrl or a.218 were injected into the dorsal hippocampus at P2. Inhibition of miR-218 had no effect on GABAergic synapse number in the CA3 pyramidal layer as measured by VGAT and gephyrin puncta density at P8 (*Figure 6A–B*). There was also no effect on either amplitude or frequency of miniature excitatory and inhibitory postsynaptic currents (mEPSCs and mIPSCs) in CA3 pyramidal neurons at P14-P16 (*Figure 6C–D*). We then asked if, instead, there was a structural defect causing changes in connectivity. We measured spine density at P11 by co-injecting antagomiRs with a GFP-expressing pseudorabies virus to sparsely label neurons. We saw no effect on spine density in CA3 pyramidal neurons. In contrast, early miR-218 inhibition significantly reduced spine density in CA1 pyramidal neurons, the major postsynaptic targets of CA3 (*Figure 6E–F*). The reduction in spines was most prominent among mushroom-type spines. We then tested whether this reduction in spines was long-lasting by co-injection of a.218 or a.Ctrl with AAV-smFLAG at P2 and measuring spine density at P40-P45. Early, transient inhibition of miR-218 produced a long-lasting reduction in spines on CA1 pyramidal neurons (*Figure 6E–F*). The phenotypic differences between CA1 and CA3 PNs could be due to different levels of miR-218 (*Figure 4A*) and/or different miR-218-regulated gene programs in each cell type. The reduction in spine density in CA1 could also reflect a homeostatic compensatory response to the increased activity levels in CA3 (*Figure 1E–I*).

## MiR-218 regulates intrinsic excitability

Because we did not observe changes in synaptic input to CA3 PNs, we measured intrinsic properties of these neurons. Previous reports demonstrated that miR-218 regulates neuronal excitability in spinal cord motor neurons and cultured neurons (*Amin et al., 2015*; *Reichenstein et al., 2019*). Further, a subset of predicted miR-218 targets directly affect intrinsic membrane properties. *Kcnd2* and *Kcnh1* encode for voltage-gated potassium channels (Kv4.2 and Kv10.1, respectively), major regulators of membrane excitability (*Gross et al., 2016*; *Mortensen et al., 2015*; *Simons et al., 2015*). MiR-218 is also predicted to target *Kcnip3*, which interacts with voltage-gated potassium channels and regulates their activity (*Grillo et al., 2019*). We measured intrinsic properties of CA3 pyramidal neurons at P5-P6. We found that miR-218 regulates neuronal excitability, as a.218 increased membrane resistance and decreased the rheobase current (the minimal current needed to elicit an action potential) compared to a.Ctrl (*Figure 7A–D*). Thus miR-218 inhibition increases the excitability of CA3 PNs independent of synaptic input, consistent with the increased network activity described above.

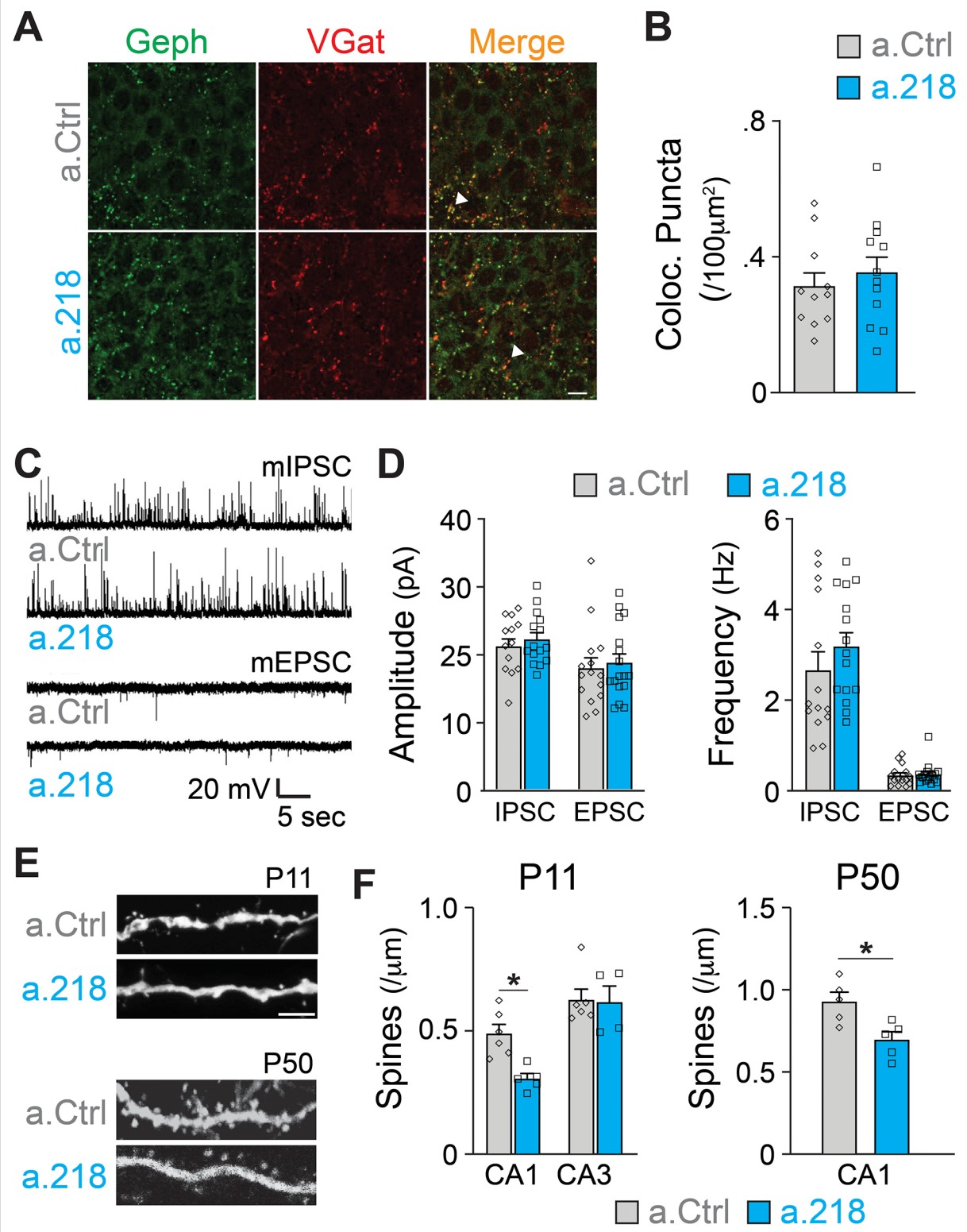

**Figure 6.** MiR-218 inhibition induces selective structural changes. (**A**) Immunostaining of VGAT (red) and Gephryn (green) in the stratum pyramidale of the hippocampal CA3 region at P8. Scale bar = 10 μm. (**B**) Quantification of co-localized puncta (yellow). N for a.Ctrl=11 sections/3 mice, for a.218=12 sections/3 mice. (**C**) Traces of mIPSC and mEPSC recordings in CA3 PNs at P14-P16. (**D**) Quantification of mIPSC and mEPSC amplitude and frequency in a.218 vs a.Ctrl mice. N=14–17 cells from 4 to 6 mice/group C. (**E–F**) Analysis of dendritic spines on secondary proximal dendrites in CA1 and CA3 at P11 and P50. a.218 induces a decrease in spines in CA1 that persists in the adult. Scale bar = 1 μm. N=5–6 mice/group. Bar graphs: mean ± SEM; *p<0.05. Statistics were calculated using linear mixed models accounting for animals (**B and D**), and unpaired t-tests (**F**).

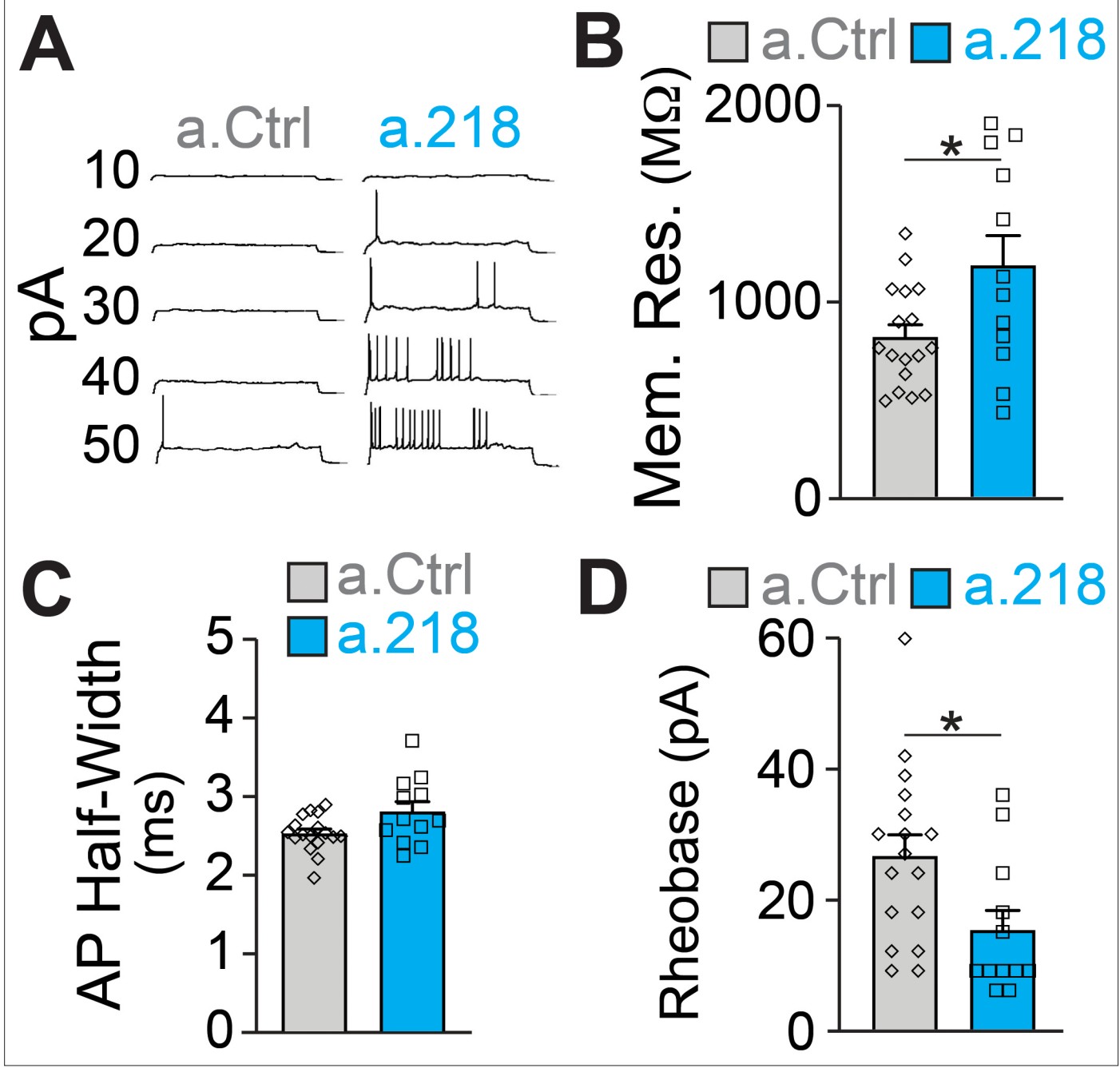

**Figure 7.** MiR-218 inhibition induces changes in neuronal intrinsic properties. (**A**) Traces of whole-cell current clamp recordings of CA3 PNs in acute slices from a.Ctrl (left) and a.218 (right) mice showing spiking in response to current injection. a.218 increases membrane resistance (**B**), has no effect on action potential (AP) half-width (**C**), and reduces rheobase (**D**) compared to a.Ctrl. N for a.Ctrl=17 cells/9 mice; for a.218, N=12 cells/9 mice. Bar graphs: mean ± SEM; *p<0.05. All statistics are calculated from linear mixed models accounting for animals as a random effect.

## MiR-218 regulates early depolarizing GABAergic neurotransmission

GABAergic neurotransmission is a key component of immature neuronal activity, a critical driver of circuit formation. To test if miR-218 inhibition impairs GABAergic signaling, we performed calcium imaging of acute hippocampal slices at P5, when glutamatergic synapses are immature and SEs are primarily driven by depolarizing GABA (*Ben-Ari, 2002*; *Ben-Ari et al., 1989*; *Duan et al., 2020*). Inhibition of miR-218 had a pronounced effect on both the frequency and pattern of spontaneous activity in CA3 at P5. Loss of miR-218 activity significantly decreased the frequency of SEs and total

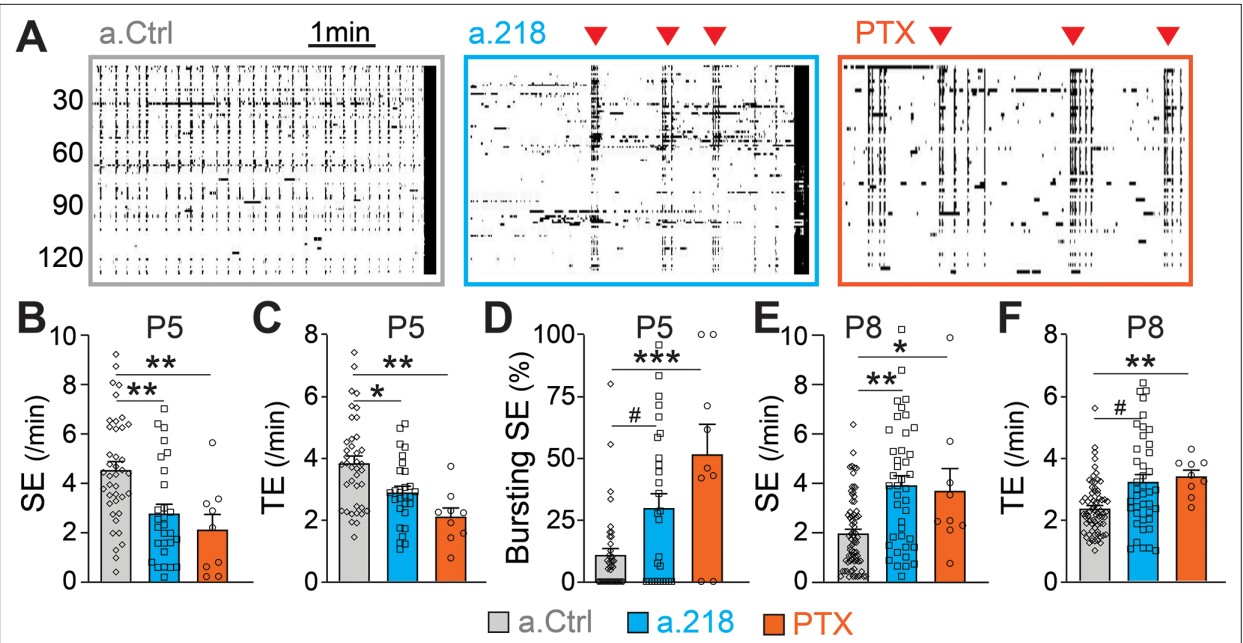

**Figure 8.** MiR-218 inhibition alters GABAergic neurotransmission at P5. (**A**) Raster plots of neuronal activity in movies from P5 a.Ctrl mice, a.218 mice, or slices from WT mice treated with picrotoxin (PTX). Each line is a cell, black dots represent calcium spikes. Black vertical lines indicate SEs. Red triangles indicate bursts of SEs. a.218 mice and PTX slices show fewer SEs (**B**) and total events (TE, **C**) that tend to occur in bursts (**D**). N=36 movies/ 5 animals (a.Ctrl), 28 movies /5 animals (a.218), 9 movies/3 animals (PTX). (**E–F**) In contrast to P5, at P8 both a.218 and PTX increase the frequency of SEs (**E**) and total events (**F**). a.218 data and comparisons in (**E–F**) are from *Figure 1F–G*. N=9 movies/3 animals (PTX). Bar graphs: mean ± SEM; #p<0.1, **p<0.01, ***p<0.001. We fit linear mixed models accounting for animal as a random effect between PTX and a.Ctrl conditions at P5 and P8 or a.Ctrl and a.218 conditions at P5.

events (*Figure 8A–C*). In addition, a.218 frequently caused SEs to occur in quick bursts, rather than at the evenly spaced intervals seen in control conditions (*Figure 8D*). Acute blockade of GABAergic signaling by bath application of picrotoxin (PTX, 50 µM) on P5 acute slices produced effects similar to miR-218 inhibition. This included a reduction in SEs and total events (*Figure 8A–C*) with an accompanying shift from regularly spaced SEs to bursting patterns (*Figure 8D*). These data support the hypothesis that impairment of GABAergic neurotransmission could contribute to the a.218-induced phenotypes. As a further test of this hypothesis, we tested acute blockade of GABA on CA3 spontaneous activity at P8, when glutamatergic neurotransmission has begun and GABA, although still depolarizing, has been shown to mediate shunting inhibition (*Staley and Mody, 1992*). Indeed, at P8, acute GABA inhibition by PTX (100 µM) increased the frequency of CA3 SEs (*Figure 8E*), and total events (*Figure 8F*), similar to the effects of miR-218 inhibition (*Figure 1*). These results indicate that miR-218 activity is critically involved in establishing early neuronal activity patterns, possibly through disruption of GABAergic signaling.

### *Mir218-2* KO in INs but not in PNs recapitulates a.218 effects on long-term hippocampal stability

MiR-218 regulates the structure and the intrinsic excitability of young PNs. In addition, miR-218 regulates GABAergic neurotransmission with broad repercussions for spontaneous neuronal activity. To understand which of these phenotypes is causing long-term deficits in network stability, we selectively deleted *Mir218-2* from PNs or INs using a *Mir218-2* cKO line (*Mir218-2*^fl/fl). We used VGluT2-Cre and VGAT-Cre mice to knockout *Mir218-2* from PNs and INs, respectively. We then tested the efficiency of the cell type-specific *Mir218-2* KO. Crossing VGluT2-Cre/Ai14 and VGAT-Cre/Ai14 lines with *Mir218-2*^fl/fl allowed us to measure miR-218 levels in these cell types using FACS followed by miRNA qPCR (*Figure 9A–B*). Compared to WT littermates, miR-218 levels in VGAT-Cre/Ai14/*Mir218-2*^fl/fl mice were markedly reduced at P8. A significant reduction was also observed in VGluT2-Cre/Ai14/*Mir218-2*^fl/fl at P14. To confirm the functional KO of miR-218 in PNs and INs, we used the miR-218 sensor (*Figure 9C*).

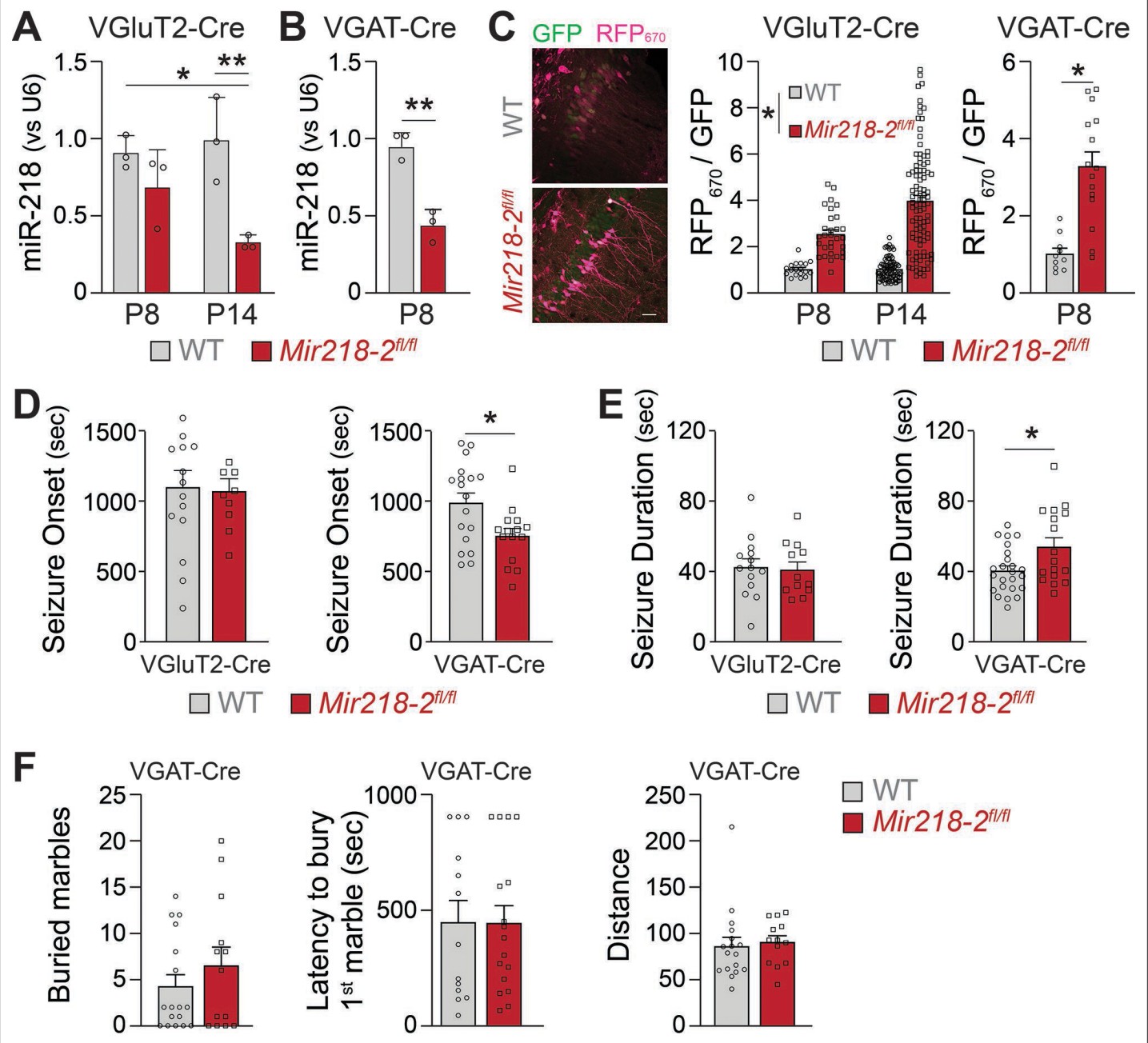

**Figure 9.** *Mir218-2* cKO in INs but not in PNs alters long-term network stability. MiR-218 was quantified in VGluT2-Cre/Ai14/*Mir218-2*[fl/fl] (**A**) and VGAT-Cre/Ai14/*Mir218-2*[fl/fl] (**B**) using FACS followed by qPCR. N=3 animals/group (**C**) Decrease in miR-218 function was confirmed with a miR-218 sensor (see **Figure 4G** for details about the sensor) for both PNs and INs. N=8–12 sections from 3 mice/group. Scale bar = 50 μm. (**D**) VGAT-Cre/Ai14/*Mir218-2*[fl/fl] mice show faster seizure onset after KA delivery, while VGluT2-Cre/Ai14/*Mir218-2*[fl/fl] show no differences. (**E**) VGAT-Cre/Ai14/*Mir218-2*[fl/fl] mice show longer seizure duration, while VGluT2-Cre/Ai14/*Mir218-2*[fl/fl] mice show no differences. N=9–18 mice/group. (**F**) Marble burying test shows no differences between VGAT-Cre/*Mir218-2*[fl/fl] mice and littermate controls. Two-way ANOVA with Tukey's multiple comparison test (**A**), unpaired t-test (**B**), linear mixed models accounting for animals as a random effect and age as a fixed effect (**C**), Mann-Whitney test (**D–F**). Bar graphs: mean ± SEM; *p<0.05, **p<0.01.

PNs in the CA1 region at P8 and P14 showed robust RFP_{670} de-repression, confirming efficient cKO of *Mir218-2*. A similar de-repression was observed in INs at P8. We then repeated the KA-induced seizure experiments in either VGluT2-Cre/*Mir218-2*[fl/fl] or VGAT-Cre/*Mir218-2*[fl/fl] mice (**Figure 9D–E**). *Mir218-2* KO in INs induced earlier seizing compared to WT littermates, phenocopying the effect seen with a.218. It also induced longer seizures, indicating a propensity for pathological activity. Somewhat surprisingly, considering the extent of phenotypes observed in PNs, *Mir218-2* KO in PNs had no effect

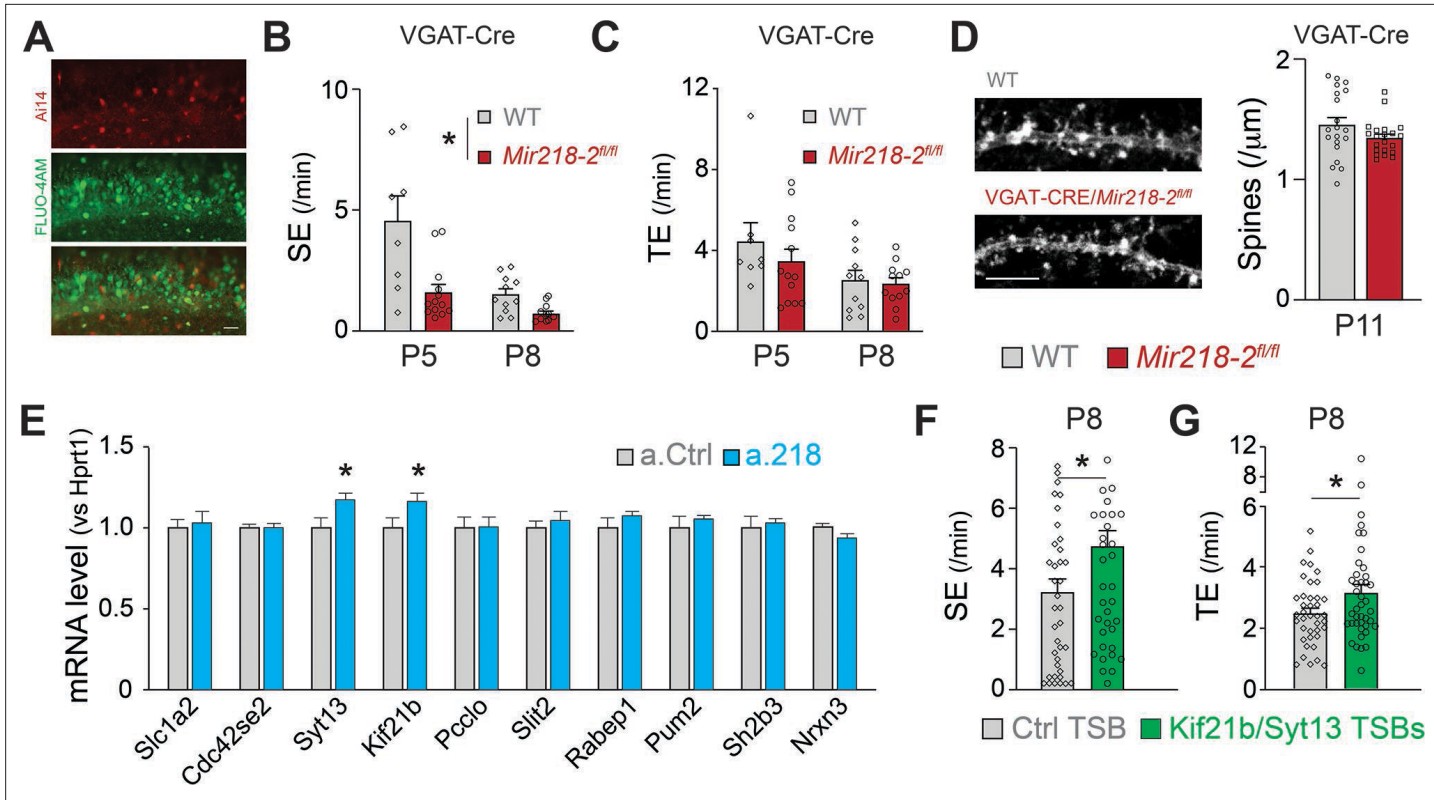

**Figure 10.** Cellular and molecular mechanisms leading to altered spontaneous network activity. (**A**) Images of VGAT-Cre/Ai14/*Mir218-2*^fl/fl slices used for calcium imaging, showing INs (top, red) in which Cre recombination occurred and PNs loaded with FLUO-4AM (middle, green). Scale bar = 50 µm. (**B**) SEs are decreased in VGAT-Cre/*Mir218-2*^fl/fl mice vs littermate controls. N=2–6 movies/3 animals per condition. (**C**) No changes in TEs. (**D**) No changes in CA1 PN spine density at P11 in VGAT-Cre/*Mir218-2*^fl/fl mice. N=4–17 dendrites/2–3 animals per condition. Scale bar = 5 µm. (**E**) *Syt13* and *Kif21b* are significantly upregulated in the hippocampi of a.218-treated mice. N=3 animals per condition. (**F–G**) *Kif21b* and *Syt13* TSBs induce an increase in SEs and TEs. N for Ctrl TSB = 39 movies/8 animals, for *Kif21b/Syt13* TSBs 41 movies/7 animals. Statistics for B, and C are derived from linear mixed models accounting for animals as a random effect and age as a fixed effect. Statistics for F, and G are derived from linear mixed models accounting for animals as random effects. Bar graphs: mean ± SEM. *p<0.05.

on KA-induced seizures. *Mir218-2* KO in PNs or in INs had no effect on EEG waveforms, suggesting that inhibition of miR-218 simultaneously in both cell types might be necessary to induce the phenotype. In addition to unstable networks, a.218 induced mild cognitive deficits (*Figure 3*). To assess if *Mir218-2* cKO in INs is sufficient to recapitulate a.218-induced behavioural defects, we performed the marble burying test in VGAT-Cre/*Mir218-2*^fl/fl mice (*Figure 9F*), but observed no significant differences with wild-type littermate controls. These experiments indicate that lack of miR-218 in INs negatively affects circuit formation, resulting in adult networks that are prone to pathological activity, but is not sufficient to fully recapitulate the long-term a.218 phenotype.

## *Mir218-2* KO in INs recapitulates aspects of the a.218 effect on early depolarizing GABAergic neurotransmission

We sought to identify the developmental mechanisms leading to long-term network instability in VGAT-Cre/*Mir218-2*^fl/fl mice. The changes in spontaneous network activity induced by PTX (*Figure 8*) suggest that GABAergic neurotransmission might be defective in the absence of miR-218. We tested if *Mir218-2* cKO in INs was sufficient to recapitulate this phenotype. We performed calcium imaging in VGAT-Cre/*Mir218-2*^fl/fl mice at P5 and P8 (*Figure 10A–C*). We observed a significant decrease in SEs in slices from the cKO mice. TEs were not changed, and we did not observe an increase in bursting SEs. The results are consistent with a large body of literature suggesting that GABA is the main driver of synchronous activity at P5. Our data suggest that miR-218 is essential for proper IN function at this stage. Interestingly, the P8 phenotype from VGAT-Cre/*Mir218-2*^fl/fl mice is opposite to what we observed with miR-218 inhibition (*Figures 1 and 8*). This suggests that miR-218 loss-of-function

simultaneously in PNs and INs induces a complex compound phenotype that is likely driven by changes occurring in PNs. To test if lower synchronous activity in VGAT-Cre/*Mir218-2*^fl/fl mice affected neuronal structure, we counted CA1 PN dendritic spines at P11 (*Figure 10D*). We found that miR-218 cKO in INs is not sufficient to induce changes in PN spines, suggesting that this phenotype is cell-autonomous or mediated by the increase in synchronous activity observed with a.218 at P8.

## Selective de-repression of the miR-218 targets *Syt13* and *Kif21b* phenocopies a.218-induced increase in synchronous activity

Our data suggest that although miR-218 plays an important role in INs, its function in PNs is critical for proper network assembly. We sought to understand the molecular mechanisms leading to the increased activity at P8. To identify *bona fide* miR-218 targets, we validated a subset of mRNAs that were up-regulated in a.218 and *Mir218-2*^+/- RNA-Seq experiments. Of the ten putative miR-218 targets tested, only *Syt13* and *Kif21b* showed significant up-regulation at P8 in hippocampi injected with a.218 at P2 (*Figure 10E*). To test if de-repression from miR-218 of these two targets alone was sufficient to phenocopy aspects of the a.218 phenotype, we used target site blockers (TSBs), as we have previously done (*Lippi et al., 2016*; *Dulcis et al., 2017*). TSBs are antisense oligonucleotides that are complementary to a specific miRNA binding site, in this case to the miR-218 binding sites on *Syt13* and *Kif21b*. We performed calcium imaging experiments at P8 in slices from mice treated with both *Syt13* and *Kif21b* TSBs or a control TSB. De-repression of the two miR-218 targets induced an increase in both SEs and TEs, phenocopying aspects of the a.218 effect. Our data indicate that maintaining proper levels of *Syt13* and *Kif21b* is necessary to keep network activity within a physiological range.

## Discussion

The early postnatal period is a critical time in the development of neuronal circuits. Spontaneous network activity plays an important role during this timeframe in shaping the properties of mature circuit function (*Ben-Ari, 2002*; *Duan et al., 2020*). The genetic programs underlying early spontaneous activity are subject to both transcriptional and post-transcriptional regulation. We have previously shown that a single miRNA can regulate multiple aspects of circuit development in early postnatal life (*Lippi et al., 2016*). In the present study, we further demonstrate that a distinct developmentally regulated miRNA, miR-218, influences postnatal hippocampal development through effects on neuronal morphology, excitability, and network activity. The long-term effects are likely due to alterations in neuronal excitability and disruption of GABAergic signaling. We further show that miR-218 is required specifically in INs for proper network development, and that miR-218 functions in part through regulation of the kinesin *Kif21b* and the synaptotagmin *Syt13*.

An increasing body of work has shown that abnormal expression of miR-218 is associated with several nervous system disorders, including major depressive disorder (*Torres-Berrío et al., 2021*; *Torres-Berrío et al., 2020*; *Torres-Berrío et al., 2017*), amyotrophic lateral sclerosis (*Reichenstein et al., 2019*), and temporal lobe epilepsy (*Kaalund et al., 2014*). Our work importantly extends these findings to show that inhibiting miR-218 during circuit development, particularly in INs, results in increased seizure susceptibility and severity, thus establishing a causal link between miR-218 levels and neurodevelopmental phenotypes.

Manipulation of miR-218 levels has repeatedly been shown to alter neuronal activity in both spinal cord and hippocampal neurons. Our findings that inhibition of miR-218 resulted in hyperactive network activity and hyperexcitable CA3 pyramidal neurons are consistent with recordings from *Mir218* KO embryonic spinal cord motor neurons (*Amin et al., 2015*). Recent data from miR-218 knockdown in cultured primary hippocampal neurons indicated that miR-218 inhibition decreased neuronal activity (*Reichenstein et al., 2019*). Additional studies with constitutive *Mir218-2* KO mice showed a 50% reduction in miR-218 in the hippocampus, cognitive deficits, decreased activity, loss of long-term potentiation and increased spine density in CA1 pyramidal neurons (*Lu et al., 2021*). Several factors could explain the discrepancies between our findings and these published results, including differences in the extent and duration of knockdown/KO, or different effects of miR-218 across developmental ages (15–21 days in vitro for cultured neurons, P30-40 for *Mir218-2* KO CA1 neurons) and brain regions (CA3 vs CA1). The regulation of a given mRNA by miR-218 is likely highly dependent upon the relative concentrations of both the target mRNA and the miR-218 (*Amin et al.,*

*2021*). It is therefore plausible that miR-218 will regulate distinct cohorts of target mRNAs in different cell types and at different developmental time points.

Prior work has shown miR-218 is expressed in both excitatory PNs and INs within the mouse nervous system (*He et al., 2012*). In this study, we observe effects of miR-218 inhibition on the intrinsic excitability of early postnatal hippocampal PNs. In many neuron types, intrinsic properties, including membrane resistance and rheobase, change extensively during early postnatal development (*Perez-García et al., 2021*). This may be due to changes in the expression and/or distribution of K+channels, some of which have been shown to be miR-218 targets (*Reichenstein et al., 2019*). Thus, it is possible that miR-218 inhibition at later stages could have different effects on intrinsic excitability of PNs than those we describe here during the early postnatal window.

While we find that inhibiting miR-218 renders PNs hyperexcitable at P5-6, the overall effect is to decrease network activity. This is likely due to a role of miR-218 in promoting GABAergic signaling. Several lines of evidence support this conclusion. First, miR-218 is expressed in a subset of INs (*He et al., 2012*). Second, a subset of genes upregulated following miR-218 inhibition are enriched in INs compared to PNs in the postnatal hippocampus. Third, the effects of miR-218 inhibition on spontaneous network activity are strikingly similar to the effects of acute GABAergic blockade at both P5 and P8. Fourth, removal of *Mir218-2* from INs, but not PNs, confers increased susceptibility to seizures in mature animals and reduces network activity at early postnatal ages. Thus miR-218 regulates GABAergic signaling in the early postnatal hippocampus to establish proper network formation. These data are consistent with work showing that GABA signaling during early development is crucial for the proper establishment of synaptic architecture and excitatory/inhibitory balance in mature circuits (*Ben-Ari et al., 2012*; *Cherubini et al., 2021*; *Oh et al., 2016*; *Salmon et al., 2020*). Removal of *Mir218-2* specifically from INs, however, only leads to a subset of the phenotypes of global miR-218 inhibition.

We confirmed two downstream targets of miR-218 in the hippocampus, the kinesin *Kif21b* and the synaptotagmin *Syt13*. *Kif21b* and *Syt13* were up-regulated in both the a.218 vs. a.Ctrl RNA-Seq and the *Mir218-2*$^{-/-}$ vs WT RNA-Seq experiments. Consistently, both genes were also highly up-regulated in *Mir218*$^{-/-}$ spinal cord motoneurons (*Amin et al., 2015*). Mutations in *Kif21b* have been shown to impair neuronal migration, axonal growth and branching, leading to neurodevelopmental disorders (*Asselin et al., 2020*; *Narayanan et al., 2022*). The vesicle trafficking protein SYT13 has been shown to regulate insulin secretion and cell-matrix adhesion in the pancreas (*Bakhti et al., 2022*; *Ofori et al., 2022*) and to promote axon growth and protect motor neurons from degeneration (*Nizzardo et al., 2020*). Overall, these findings strongly suggest that miR-218 functions in part through modulating intracellular trafficking and is required in both PNs and INs in concert to regulate intrinsic and network excitability.

## Materials and methods

### Animals

Animal experiments were conducted at the University of California, San Diego (UCSD), the Scripps Research Institute (SRI), Cincinnati Children's Hospital Medical Center (CCHMC), and at the University of Modena and Reggio Emilia (UNIMORE). C57Bl/6 J animals were all ordered from the same supplier (Envigo). Constitutive *Mir218-1*$^{-/-}$ and *Mir218-2*$^{+/-}$ mice were kindly provided by Dr. Samuel Pfaff. Conditional *Mir218-2*$^{fl/fl}$ mice were kindly provided by Dr. Gian Carlo Bellenchi. VGAT-IRES-Cre (*Slc32a1*$^{tm2(cre)Lowl}$/J, JAX strain #016962), VGluT2-IRES-Cre (*Slc17a6*$^{tm2(cre)/Lowl}$/J, JAX strain #016963), and Ai14 (B6.Cg-*Gt(ROSA)26Sor*$^{tm14(CAG-tdTomato)Hze}$/J, JAX strain #007914) mice were purchased from Jackson Laboratories.

Animals were housed under 12/12 hr light/dark cycle (07:00 AM-7:00 PM) at 23 ± 1°C with free access to food and water. All experimental procedures at UCSD, CCHMH and SRI were performed as approved by the Institutional Animal Care and Use Committees and according to the National Institutes of Health Guidelines for the Care and Use of Laboratory Animals. Behavioral and in vivo experiments at UNIMORE were conducted in accordance with the European Community Council Directive (86/609/EEC) of November 24, 1986, and approved by the ethics committee (authorization number: 37/2018PR).

## P2 Injections

Surgeries were performed under sterile conditions. General anesthesia was induced in P2 neonates with deep hypothermia alone as described (*Lippi et al., 2016*; *Lozada et al., 2012*). Briefly, the pups were placed on crushed ice covered with Saran wrap to avoid freeze damage to the skin. Injections were performed with a Nanoject (*Lozada et al., 2012*) (Drummond Scientific Company, Broomall, PA) and a beveled glass injection pipette (*Adesnik et al., 2008*). Animals received two injections in each dorsal hippocampus (see *Figure 1D* for an illustration of the procedure). The glass pipette (Flared Glass: 4.45 cm long, 90 µm diameter) was lowered 2 mm below the skull level and then retracted 1 mm; 207 nl were injected for each site. After surgery, pups were placed on a heated pad for few min until they were warm and capable of spontaneous movement. All traces of blood were gently removed from the pups to minimize chances of the dam rejecting the pups. Concentrations used were 0.5–1 µM for all locked-nucleic acid antagomiRs (Exiqon).

The antagomiRs injected had the following sequences:

 a.Ctrl: gtgtaacacgtctatacgccca
 a.127: gctcagacggatccg
 a.136: tcaaaacaaatggag
 a.218: ggttagatcaagcaca

Target site blockers (TSBs) designed to prevent miR-218 from interacting with its miRNA recognition elements (MREs) on *Syt13* and *Kif21b* were ordered from Exiqon. We used two TSBs for the two miR-218 MREs on *Syt13* and four TSBs designed against four miR-218 MREs on *Kif21b*. The six TSBs were designed by Exiqon to be combined in vivo with minimal cross-interactions. The TSBs were combined to a final concentration of 1 µM.

14–16 nt TSBs were designed against the following sequences (miR-218 seed in capitals):

*Syt13* TSB1:

tctacccttgctgtttcctgttgttagatgctgcgtgtttgtgatgccattccacataaAAGCACAAaaagtcgcatgagacatcgtg gtcacatgtctggttacactttggagcacaaaatcttacagtggtaaataaatcgttttccaatcgggttggcagcccagtgtt

*Syt13* TSB2:

tctacccttgctgtttcctgttgttagatgctgcgtgtttgtgatgccattccacataaaagcacaaaaaagtcgcatgagacatcgtg gtcacatgtctggttacactttggAGCACAAaatcttacagtggtaaataaatcgttttccaatcgggttggcagcccagtgtt

*Kif21b* TSB1:

aggtacatagtcaggggcaggaccccagctccactcattgcacaagcatccttcttgtcggAAGCACAAactggttgtaatttg agcacaatagcaggtctctgggtagacagtatgagcaaacttggcaccaggagatctttcccagatgggct

*Kif21b* TSB2:

aggtacatagtcaggggcaggaccccagctccactcattgcacaagcatccttcttgtcggaagcacaaactggttgtaatttg AGCACAAtagcaggtctctgggtagacagtatgagcaaacttggcaccaggagatctttcccagatgggct

*Kif21b* TSB3:

accgttgctccagctgcaggcctggaatggtcttaggctgtgtggcttgggcaaagggcagcttaAAGCACAAcaagga gaagatgctgtcttgctgaaggattcttctgctagccattgtggtgg

*Kif21b* TSB4:

cagactgcagggtctcgtttggctcatgcactctcctgtgacgtagtAGCACAAggtgtgtatatgttttgtacctctgc  tgacaac tgtacatagtgtatgaaagttatttaagcctcatgctgtacatttct

For the calcium imaging experiments in P40 mice, 5 mM antagomiRs were mixed at room temperature with GcaMP6f adeno-associated viral particles (UPenn viral core, AAV1.Syn.GcaMP6f.WPRE.SV40) at a 1:1 ratio before the injection. Male mice were injected at P2 as described, weaned at 4 weeks of age, and housed up to 6 per cage. For the analysis of spines at P11, a glycoprotein-deleted variant of the SAD-B19 strain of rabies virus encoding eGFP (SADΔG-eGFP) was mixed with antagomiRs at a 1:10 ratio. Such a low titer was necessary to achieve sparse labeling of both CA1 and CA3 pyramidal neurons. The virus was kindly provided by Byungkook Lim. For the analysis of spines at P50, AAV (pAAV.CAG.Ruby2sm-FLAG.WPRE.SV40) was mixed with antagomiRs (1:1 ratio), and then injected at P2 as described. Mice were weaned at 4 weeks of age and perfused at P40-P45.

## Calcium imaging

### P5 and P8 calcium imaging

Calcium Imaging at P5 and P8 was performed as described (*Lippi et al., 2016*). Briefly, C57B/L6J P8 male mice were anesthetized with Ketamine/Xylazine and sacrificed by decapitation. Acute coronal slices (300 µm) were prepared using a Leica vibratome with ice-cold oxygenated slicing solution as described above. We obtained 3–4 coronal slices containing both dorsal hippocampi per animal. To load the calcium indicator, slices were transferred into a sealed 1.5 ml tube containing 5 µM Fluo-4AM (Molecular Probes, 50% DMSO, 50% Pluronic Acid) diluted in oxygenated modified artificial cerebrospinal fluid (mACSF) at 35 °C, for 30 min in the dark. The mACSF (also used as imaging solution) contained (in mM): 126 NaCl, 3.5 KCl, 1.2 NaH2PO4, 26 NaHCO3, 1.0 MgCl2, 2.0 CaCl2, and 10 glucose, pH 7.4. Slices were then dissected along the midline and transferred to a slice holder where they were maintained at room temperature in the dark for an additional 1 hr before imaging. Experiments were performed in a custom-made chamber that allowed constant perfusion with aerated mACSF at 1–2 ml/min. A Leica SP5 multiphoton microscope (Leica Microsystems, Germany) with a low-magnification (10 X) 36 water immersion objective was used for imaging. Bi-directional scanning was used at a 400 Hz imaging rate, with a time resolution of 328ms. The image field was typically 620x310 µm. Follow-up experiments were performed with a Nikon A1R Multiphoton system using a 16x CF175 LWD objective, N.A. 0.8, 488nmm laser with the same frame rate as described previously. After 5–10 min of imaging, 5 µL of potassium chloride (KCl, 1 M) was added to the aCSF immediately above the slice to depolarize neurons and detect healthy cells. For calcium imaging at P8, n=72 movies from 26 animals, 33 movies from 7 animals, 23 movies from 5 animals, 41 movies from 11 animals, for a.Ctrl, a.127, a.136, and a.218, respectively. From P5, n=36 movies from 5 animals and 28 movies from 5 animals for a.Ctrl and a.218, respectively. For experiments with picrotoxin (PTX), PTX was added to the aCSF (final concentration 50–100 µM) being perfused through the imaging chamber. For P5 movies with PTX, n=9 movies from 3 animals, for P8, 9 movies from 3 animals.

### P40 calcium imaging

Briefly, mice are anesthetized with Ketamine/Xylazine and perfused through the heart with 10 ml of ice-cold oxygenated slicing solution containing (in mM): 93 N-methyl-D-glucamine (NMDG), 20 Hepes, 2.5 KCl, 1.2 NaH2PO4, 30 NaHCO3, 25 glucose, 10 MgSO4, 0.5 CaCl2, 5 sodium ascorbate, 3 sodium pyruvate, 2 thiourea, pH 7.4. Acute coronal slices (300 µm) were prepared using ice-cold oxygenated cutting solution. On average we obtained 5–6 coronal slices containing both dorsal hippocampi per animal. Slices were allowed to recover for 15 min at 35 °C and then were transferred to oxygenated mACSF at room temperature for another 1 hr. Slices were then dissected along the midline and transferred to a slice holder where they were maintained at room temperature in the dark for an additional 1 hr before imaging.

### Analysis

Movies were analyzed as described (*Lippi et al., 2016*). All analyzers were blinded to the conditions of the movies. Briefly, raw movies were aligned with the ImageJ StackReg plugin. A custom-made analysis pipeline was developed using Matlab (Math-Works; RRID:SCR_001622) and the imaging computational microscope (ICM) (*Frady et al., 2016*). Cells were identified automatically using principal component analysis and independent component analysis (*Mukamel et al., 2009*) (PCA; ICA), and were manually verified, and only cells that responded to the KCl-induced depolarization at the end of the imaging were retained. Cells that had very slow calcium responses (>5 s) were excluded, on the basis that they were most likely glia. The calcium signal trace for each cell was obtained by averaging the fluorescence within that ROI as a function of time.

We adjusted for photo-bleaching-induced decreases in baseline fluorescence as follows. First, we first smoothed the data by applying a band-pass filter (cutting high-frequency oscillations); then we determined a "support line" (SL), precisely a cubic curve underneath the smoothed series whose distance from the series was minimum. The filtered series was the difference between the series and SL. The KCl-induced depolarization at the end of the movie was not included to compute SL. A manual threshold (normally ≥3 fold the standard deviation of baseline fluorescence or F0) was used to identify calcium events. The synchronized events (SEs) were detected by selecting the principal component (PC) that corresponded manually using ICM, which was typically the first or second PC. Because these

events were correlated and accounted for a lot of variance in the data, they were easily extracted by PCA. The number of SEs was determined by comparing raster plots of all calcium events extracted with ROIs with the corresponding PC that signifies the event. The number of cells participating in each SE was determined by counting how many cells, out of all viable cells (those which responded to KCl), had calcium events in the time window of 984ms before and after the peak of the SE. Asynchronous events were all events that were not included in the time window of each SE.

Data from each movie were then transferred to Excel (Microsoft) for consolidation, and statistical analysis was performed in Prism (GraphPad). The ROUT method was used for detecting outliers. At P8, outliers were detected based on the number of SEs. These movies were removed for all further analysis. For a.Ctrl three outliers were detected and removed, for a.127, two outliers were removed, and for a.136 four outliers were removed. No outliers were detected among the a.218 movies. At P40, three outliers were removed from a.218 movies, whereas none were detected in the a.Ctrl data. One outlier was detected and removed from a.218 P5 recordings. For P5 movies, the percent of SEs occurring in bursts was determined from the PCA trace. A burst of SEs was determined by multiple SEs occurring within a time interval such that the baseline of the PCA trace did not return to baseline in between SEs. The percent of total SEs in each movie occurring within bursts was manually counted.

### Lentiviral miRNA sensor

The miR-218 lentiviral sensor was created from the lentiviral miR-101b tracer, a generous gift from Dr. Fred Gage (*Han et al., 2016*; *Lippi et al., 2016*). The 6 miR-101b sites in the 3' UTR of $iRFP_{670}$ were replaced with 8 copies of the miR-218 miRNA recognition element (MRE). A control sensor (not sensitive to any miRs) was cloned with no known miRNA sequences in the $iRFP_{670}$ 3' UTR. Lentivirus was packaged as described (*Boyden et al., 2005*; *Han et al., 2009*).

### Testing antagomiR efficiency

Purified and concentrated lentiviral sensor was combined with a.Ctrl or a.218 (final concentration of antagomiR 0.5 μM), and injected into P2 animals. At P8, P15, or P26, animals were perfused with PBS followed by 4% paraformaldehyde, and brains were removed and postfixed overnight in 4% PFA. Brains were cryoprotected in 30% sucrose in PBS until they sank. Sections (40 μm) were obtained on a sliding microtome (Leica), mounted onto slides, and coverslipped with VectaShield. Slides were imaged on a Leica SP5 laser scanning confocal microscope. Virally-infected cells were identified by GFP fluorescence, and Z-stacks (at 2 μm steps) were taken through the entire section using a 20 X oil immersion objective. Acquisition settings (laser power, gain, offset) for GFP and $iRFP_{670}$ remained identical for all conditions. Max projections of z-stacks were analyzed in ImageJ. ROIs were drawn using the GFP channel, and the average fluorescence intensity was measured in both the GFP and $iRFP_{670}$ channels. The $iRFP_{670}$/GFP ratios for all GFP + neurons within a section were averaged to give one value for each section. To control for age-related variation in expression from the promoters, the ratio of $iRFP_{670}$ to GFP fluorescence in the miR-218 sensor was normalized to the average $iRFP_{670}$/GFP ratio of the control sensor with the same AntagomiR (a.Ctrl or a.218) at each age.

### Testing Mir218-2 cKO efficiency

Lentivirus was packaged as described (*Han et al., 2009*; *Boyden et al., 2005*). Purified lentiviral sensor was injected into P2 VGluT2 /or VGAT-Cre/Ai14/*Mir218-2*fl/fl and VGluT2 /or VGAT-Cre/Ai14/*Mir218-2*+/+ mice. At P8 or P14, the mice were transcardially perfused with PBS followed by 4% PFA, brains were removed and postfixed overnight in 4% PFA. Brains were cryoprotected in 30% sucrose in PBS until they sank. Coronal sections (20 μm) were obtained on a Leica CM1860 cryostat, mounted onto slides, and coverslipped with ProLong Antifade Diamond (Invitrogen P36970). Images were acquired on a Nikon Instruments A1 confocal laser microscope. Virally-infected cells were identified by GFP fluorescence, and Z-stacks (at 1.5 μm steps) were acquired through the entire section using a 20 X dry objective. Acquisition settings (laser power, gain, offset) for GFP and $iRFP_{670}$ remained identical for all conditions. Max projections of z-stacks were analyzed in ImageJ. ROIs were drawn using the GFP channel, and the mean fluorescence intensity was measured in both the GFP and $iRFP_{670}$ channels. VGluT2 +or VGAT + cells were identified by Ai14. The ratio of $iRFP_{670}$ to GFP fluorescence was compared between the conditions.

## VGAT, Gephyrin immunohistochemistry and quantification

For VGAT and gephyrin immunohistochemistry, animals were injected with either a.Ctrl or a.218 at P2. At P9 animals were anesthetized with Ketamine/Xylazine, intracardially perfused with cold PBS, followed by 4% PFA. Brains were removed and post-fixed overnight in 4% PFA at 4 C, then placed in 30% sucrose in PBS at 4 ° C until they sank. Coronal sections (30 μm thick) were obtained using a sliding microtome and were stored in PBS with 1% sodium azide at 4 ° C.

For antigen retrieval, free floating sections were incubated 30 min in pre-warmed sodium citrate buffer (10 mM) at 85–95° C in a water bath. Sections were cooled to room temperature in the sodium citrate buffer, then washed in PBS (2x5 min). Sections were incubated in blocking solution (10% normal donkey serum (NDS) in PBS with 0.25% TX-100) for 45 min at room temperature on a shaker. Sections were then incubated in primary antibody cocktail – Mouse anti-VGAT (Synaptic Systems, 131 011, 1:500), RmAB anti-gephyrin (Synaptic Systems, 147 008, 1:500) overnight at 4 ° C. Sections were then washed 3x5 min in PBS at room temperature and incubated in secondary antibody solution (Dk anti Rb AF 488, 1:500, Jackson ImmunoResearch; Gt anti Ms AF 633, 1:500, ThermoFisher) for 2 hr at room temp. Sections were then washed 3x5 min, mounted on superfrost slides, coverslipped with Vectashield with DAPI, sealed with nail polish, and dried for several days at 4 ° C.

Images taken with 63 X oil immersion objective on a Leica SP5 Confocal microscope. Five-μm-thick Z-stacks were acquired with optical section thickness of 0.35 μm from CA3c pyramidal layer, dorsal to the pyramidal layer, with averaging of 2 frames per optical section. Images were acquired with 400 Hz acquisition speed, 1024x1024 resolution (for a 246.03 μm x 246.03 μm image), pinhole at 95.44 μm. Quantification of colocalized VGAT and gephyrin puncta was performed in ImageJ using the Puncta Analyzer plugin (*Ippolito and Eroglu, 2010*).

## Electrophysiology

### P14-P16 mEPSC/mIPSC recordings

Coronal slices were prepared as described above for calcium imaging. Electrophysiological recordings were acquired at room temperature using a Multiclamp 700B amplifier, digitized with Digidata 1440 A, and Clampex 10.0. Miniature synaptic events were acquired in the presence of 1 μM tetrodotoxin aCSF using a Cesium-based internal pipette solution (in mM: 130 CsMeS, 3 CsCl, 1 EGTA, 10 HEPES, 2 Na-ATP, 0.3 Na$_2$-GTP; pH 7.3 OH, 280–290 mOsm). mEPSCs were acquired by clamping at –70 mV, and mIPSCs were acquired at 0 mV. mEPSCs and mIPSCs were analyzed using Clampfit 7.0 and MiniAnalysis software by experiments blinded to the conditions.

### P5-P6 Intrinsic excitability recordings

CA3 neurons were patched using an internal solution containing (in mM): 140 K-gluconate, 5 NaCl, 10 HEPES, 1 EGTA, 2 Mg-ATP, 0.3 Na$_2$-GTP, 2 MgCl$_2$. Slices were perfused with aCSF containing (in mM): D-APV, CNQX, and Gabazine to block synaptic activity. Input resistance was measured with 5 sweeps of a –8 pA, 1000ms long pulse. The 5 sweeps were averaged, and the voltage difference between steady state and baseline was calculated and divided by the current. Rheobase was measured from a series of 5 s current injections, starting with 10 pA, and increasing by 5 pA each sweep for 15–20 sweeps. The rheobase was the minimum current injection required to elicit an AP. Recordings in which access resistance changed by >15% over the course of the recording were discarded.

## Spine quantification

### Spine density at P11

Pseudorabies GFP was co-injected with either a.Ctrl or a.218 at P2. Animals were perfused with 4% PFA at P11. Forty-μm-thick coronal sections were imaged on Leica SP5 laser scanning confocal with a 63 X objective and 4 X digital zoom. Multiple dendrites per section were imaged from four to five sections per animal, with five animals for both a.Ctrl and a.218. Spines were counted in ImageJ with an observer blind to experimental condition. Spine densities were averaged for each brain section and then subsequently for each animal, resulting in n=5 for each condition.

### Spine density at P40-45

AAV.CAG.Flex.GFPsm_myc.WPRE.SV40 (*Viswanathan et al., 2015*) was injected with a.Ctrl or a.218 at P2. The mice were transcardially perfused with PBS followed by 4% PFA between P40-45. Coronal sections (40 µm) were obtained on a Leica VT1000S vibratome. Immunostaining was performed free-floating in 24-well plates. Sections were blocked and permeabilized in 5% Normal Donkey Serum and 0.5% Triton-X 100 in PBS. Sections were then incubated in goat anti-FLAG (Novus NB600-344, 1:2000) at 4 °C overnight, washed three times for 5 min each with PBS, incubated with donkey anti-Goat AF 488 (Invitrogen) at room temperature for 2 hr, washed three times for 5 min each with PBS, mounted onto slides, and coverslipped with DAPI Fluoromount-G (Southern Biotech 0100–20). Images were acquired on a Nikon Instruments C2 +confocal laser microscope. Z-stacks (at 0.13 µm steps) were acquired using a 60 X oil immersion objective with an additional 2.39 X optical zoom. Dendritic spine density was quantified using the NIS-Elements software on max projection images.

## Quantification of miRNA levels with qPCR

miRNA levels from embryonic and early postnatal hippocampi were measured as described (*Lippi et al., 2016*). Briefly, mice were rapidly decapitated and brains were dissected on ice pads. The dorsal hippocampus was isolated, immersed in Qiazol (Qiagen) and immediately homogenized by passing through a syringe with a 23-gauge needle. RNA was isolated using miRNeasy Micro kit (Qiagen) following the manufacturer's protocol. Levels of miRNA expression were assessed with TaqMan microRNA assays (Applied Biosystems). The choice of a proper housekeeping gene for quantification of small RNAs across developmental stages is critical. These are times in which the brain is rapidly growing through morphogenesis and the RNA content per neuron increases significantly. We have tested five housekeeping gene candidates (Thermofisher commercial names in parentheses): *Rnu6* (U6), *Snord65* (snoRNA135), *Snord68* (snoRNA202), *Snord70* (snoRNA234), *Snord45b* (snoRNA412). The up- or down-regulation of the miRNA tested in *Figure 1B* changed significantly depending on which housekeeping gene was chosen. We opted for snoRNA412 (gene name *Snord45b*) because the changes observed were close to the average of all other housekeeping genes. It was also the closest to the changes observed when the total RNA levels/mg of tissue was used for normalization. Real-Time PCR analysis was conducted on a Light Cycler 480 system (Roche), and the data were processed and analyzed using the comparative ΔCT method (*Livak and Schmittgen, 2001*).

## FACS isolation of specific cell types

P8 VGAT-IRES-Cre/Ai14, VGluT2-IRES-Cre/Ai14, P8 or P14 VGluT2 /or VGAT-Cre/Ai14/*Mir218-2*[fl/fl] and VGluT2 /or VGAT-Cre/Ai14/*Mir218-2*[+/+] mice were rapidly decapitated and brains removed and placed in Hibernate A without Ca2+ (Brainbits #HACA500), supplemented with 0.132 M trehalose and synaptic blockers (AP-V 50 µM, CNQX 20 µM). Dorsal hippocampi were dissected and diced into small pieces with a sterile scalpel. Tissue was then transferred to 1 mL tubes containing papain solution pre-warmed to 37 ° C (Worthington Biochemical LS003119, 2 mg/mL in Hibernate A without Ca2+ (Brainbits #HACA500) supplemented with 0.132 M trehalose, RPI T82000-100). Samples were incubated at 37 ° C for 60 min with mixing by inverting 4–5 times every 5 min. 100 µL of Dnase I solution (Worthington Biochemical LK003172, 1 mg/ml in Hibernate-A with 0.132 M trehalose) was added and tissue incubated 5 min at 37 ° C. Tissue was pelleted and washed (500 rcf, 2 min) twice with 400 µL Hibernate A low fluorescence (Brainbits # HALF) supplemented with 0.132 M trehalose. Tissue was dissociated by repeated rounds of trituration through fire-polished glass pipettes of 0.8, 0.4, 0.2 mm diameter. Between each round, tubes were placed on ice for 2 min, and the supernatant was removed. 400 µL of Hibernate A low fluorescence was then added for the next round of trituration. Combined supernatants were filtered through a 70 µm cell strainer (Corning #352350). Cells were pelleted by centrifugation at 4 ° C, 500 rcf for 10 min. The pellet was resuspended in 500 µL of PBS with 3% FBS and DAPI (1 µg/mL) for FACS. FACS control samples were prepared from animals lacking Ai14 and from samples without DAPI. TdTomato+/DAPI(-) cells were sorted using a 100 µm nozzle and collected into 1.5 mL tubes. Collected cells were concentrated by centrifuging at 4 ° C, 500 rcf for 10 min. The supernatant was placed in a new tube and spun at 3000 rcf for 2 min at 4 ° C. The two pellets were combined by resuspending in 750 µL of TRIzol LS and frozen on dry ice.

## RNA sequencing

### P12 small RNA-Seq

MiRNA analysis was performed in collaboration with the Yeo lab at UCSD as described (*Zisoulis et al., 2010*). Total RNA was extracted from both hippocampi of 6 P12 C57BL/6 J male mice using the miRNeasy mini kit (Qiagen). The sequencing pipeline is based on the Illumina's Small RNA Digital Gene Expression v1.5 protocol with minor modifications to include indexing and multiplexing of RNA samples. Small RNA reads were mapped to the mouse mm9 genome using Bowtie aligner (m=20, n=1, l=18, --best; RRID:SCR_005476) and then to UCSC smRNA and mirBase16 annotations (18 bp minimum; RRID:SCR_003152) to assign miRNA identity. Lastly, to account for differences in amplification efficiency due to barcoding, the reads for each miRNA were expressed as percentage of the total reads for that sample and the percentage averaged across the 6 animals to produce the pie shown in *Figure 1A*.

### P8 RNA-Seq and small-RNA-Seq in a.Ctrl vs a.218, Mir218-2$^{+/-}$ mice

P8 mice injected with either a.218 or a.Ctrl at P2, and P15 C57BL/6 and *Mir218-1*$^{-/-}$; *Mir218-2*$^{+/-}$ mice were rapidly decapitated and brains were dissected on ice pads. The dorsal hippocampus was isolated, immersed in Qiazol (Qiagen) and immediately homogenized by passing through a syringe with a 23-gauge needle. Samples were snap-frozen on dry ice. Both miRNA and mRNA sequencing were performed as described (*Kobayashi et al., 2019*; *Moore et al., 2015*). For miRNa-Seq, 100–250 µg of TRIzol extracted, DnaseI (Promega) treated total RNA were used to generate barcoded small RNA libraries using the Truseq small RNA protocol (Illumina). PCR amplified cDNA libraries were run on 6% Novex TBE gels (Illumina), and fragments running between 110–160 bp markers were gel-extracted for subsequent purification. Multiplexed libraries were sequenced as 69 bp single-end (SE) read runs on Illumina MiSeq. For RNA-Seq, 250–500 ng total RNA were rRNA-depleted (Ribo-Zero, Illumina), and barcoded libraries were prepared with the Truseq protocol (Illumina). Multiplexed libraries were sequenced as 125 bp paired-end (PE) read runs on HiSeq-2000 (Illumina). Data processing and analysis were carried out using the Galaxy suite of bioinformatic tools (*Goecks et al., 2010*) and publicly available in-house tools. All statistical analyses were carried out in R using Bioconductor packages (*Gentleman et al., 2004*). Enrichment for miRNA seed sequences in upregulated genes was performed with enrichMIR (https://ethz-ins.org/enrichMiR/), comparing 138 genes with logFC >0 and p-value <0.05 in a.218 vs a.Ctrl as the input compared to a background the list of all genes considered for differential expression (6785 genes). Enrichment for miRNA seed sequences in upregulated genes in miR-218 hemizygous vs wt mice was performed with enrichMIR, using 1017 genes with logFC >0 and p-value <0.05 compared to a background list of 11,115 genes considered for differential expression.

## GO analysis

Up-regulated and down-regulated genes were considered separately for gene ontology (GO) analysis. Genes with a logFC >0.1 for up-regulated (or <–0.1 for down-regulated) and a p-value of <0.2 were selected. GO analyses were performed using the Panther statistical overrepresentation test (https://www.pantherdb.org/).

## Comparison to single-cell RNA seq data

Single-cell RNA seq data are taken from *Rosenberg et al., 2018*, Table S5. This table lists the average expression of each gene in each cluster as TPM +1. Pseudocounts were subtracted and genes with no expression in any cluster were removed. Only protein-coding genes were considered. Enrichment of the up-regulated genes from a.218 vs a.Ctrl RNA-Seq (138 genes with p-value <0.05) was calculated for each single-cell cluster using the hypergeometric test, and p-values were adjusted using the Benjamini-Hochberg method.

## Electrode implantation and electroencephalography

6–8 week old male a.218 and a.Ctrl mice or VGluT2 /or VGAT-Cre/Ai14/*Mir218-2*$^{fl/fl}$ and VGluT2 / or VGAT-Cre/Ai14/*Mir218-2*$^{+/+}$ mice were implanted with wireless transmitters for EEG monitoring *Castro et al., 2012*; *Chang et al., 2011*; *Williams et al., 2006* following a method described by *Tse et al., 2014* and as per our previously published protocol (*Tiwari et al., 2020*). Briefly, mice were anesthetized with 4% isoflurane in medical grade oxygen in a closed chamber, and then

maintained at 0.7–1.5% isoflurane throughout the surgery and monitored for pattern of respiration. Single channel wireless EEG transmitters, (TA11ETA-F10, Data Science International (DSI), St, Paul, MN, USA) were used. The head was first shaved and disinfected using Dermachlor (2% Chlorhexidine) and carprofen (5 mg/kg) was administered subcutaneously. The skull was exposed by making an incision along the midline. Dorsoventral coordinates were measured from the bregma, and bilateral holes were drilled at AP = –2.5 mm, L = ± 2.0 mm. 1 mm length of each of the leads of the transmitter was carefully inserted into the cortex and sealed using GLUture (Abbott Laboratories, IL, USA). The wireless transmitter was then positioned subcutaneously by forming a pocket behind the neck. A screw was attached to the back of the skull and a slurry of dental cement (Patterson Dental, OH, USA) was applied to secure the assembly and the cement was allowed to dry. The open skin was closed using surgical sutures (Covidien, MN, USA) and later sealed with GLUture. Post-surgery, the mice were injected with 1 ml saline, placed on a heating pad, and monitored up until recovery.

## Video-EEG recording

Post recovery, the mice were housed in individual static cages placed on wireless receiver plates (RPC1; DSI). DATAQUEST A.R.T software was used for recording of EEG data received from the telemetry system. EEG frequency was recorded between 1 and 200 Hz and video was continuously recorded (Axis 221, Axis communication) in parallel. Three days post-surgery the mice were treated with kainic acid to measure seizure susceptibility (see below).

## Kainic acid injection and EEG analysis

Mice were injected intraperitoneally with 15 mg/kg kainic acid (2 mg/ml solution in sterile Ringers) and returned immediately to the recording platform to measure seizure onset. After 90 min of EEG-video recording, seizures were terminated by a subcutaneous injection of diazepam (15 mg/kg). EEG data were analyzed using *NeuroScore* software (DSI, MN, USA). Seizure onset was manually scored and defined as a behavioral seizure in the recorded video accompanied by a twofold change in the amplitude and frequency of baseline EEG signal. For power analysis, the raw EEG signal was exported into 10 s epochs and subjected to Fast Fourier Transform (FFT) to generate power bands as per *Tse et al., 2014*. Times of excessive moving or grooming were excluded from the analysis. Total EEG power across all frequencies (24–80 Hz) were analyzed, and cumulative EEG power within the 90-min periods were reported. Mice with EKG signals were eliminated from the EEG power analysis, resulting in N=4 a.Ctrl mice and 6 a.218 mice.

## Behavioral Experiments

Following a.Ctrl or a.218 injection at P2, mice were returned to their home cages for maternal care until weaning (P24). Mice were kept in a conventional animal facility with controlled temperature (20–24 °C) and humidity (60%), on a light/dark cycle of 12 hr (hr) and housed in standard cage (6 mice/cage) in the absence of physical and structural environmental enrichments; food and water were available ad libitum. After reaching 2 months of age, mice were subjected to a battery of behavioral tests. All animals were encoded by numbers to avoid biases and the same cohort of mice was subjected to different behavioral tests at different ages. Mouse behavior was video tracked and automatically analyzed by ANY-maze software (Stoelting) while stereotypies were manually counted. For all tests, behavioral equipment was thoroughly wiped with 70% ethanol after each animal to avoid olfactory cues.

### Elevated plus maze

The elevated plus maze (EPM) test was performed at 2 months of age on 16 a.Ctrl (5 females, 11 males) and 16 a.218 (8 females, 8 males) animals to assess anxiety. Mice were placed in the central area (9x9 cm) of a plus-shaped apparatus with two opened (40x9 cm) and two closed (40x9 x 15 cm) arms, lifted 70 cm above the ground. During the five-minute test, the number of transitions between open and closed arms and center virtual division (line crossing), total distance travelled and the percentage of time spent in the open arms were analyzed.

## Open field

Locomotor activity, exploration and anxiety were assessed in the open field (OF) test on 16 a.Ctrl (5 females, 11 males) and 18 a.218 (10 females, 8 males) mice, at 2 and 6 months of age. Animals were placed in the center of a large (50x50 x 50 cm) chamber with dark walls and allowed to explore freely for 10 min. The apparatus was virtually divided in 3 subzones, namely the center (30x30 cm), the corners (10x10 cm) and the periphery (the rest of the arena). Total distance travelled, number of line crossings, total time immobile, and percentage of time spent in each of the 3 areas were recorded. Additionally, chamber occupancy of each group was displayed and analyzed through a heatmap of the area covered by the animal's body.

## Y-maze

The Y-maze test was performed to evaluate working and spatial memory, at 2 months old, on 16 a.Ctrl (5 females, 11 males) and 18 a.218 (10 females, 8 males) mice. Animals were placed in the center of a Y-shaped maze with 3 arms (A, B and C, each 22x10 x 7 cm) and allowed to explore freely for 8 min. The number of line crossings between the center and one of the three arms and the total distance travelled by the animal were recorded. Additionally, the percentage of correct spontaneous alternations, calculated as the number of entrances into an arm not explored in the previous two entrances divided by total entrances minus two, was evaluated as an index of working memory integrity.

## Morris water maze

The Morris water maze (MWM) test was used to assess mouse spatial learning and memory (**Daini et al., 2021**). Six-month-old animals (16 a.Ctrl (5 females, 11 males) and 18 a.218 (10 females, 8 males)) were placed in a circular pool (90 cm diameter) filled with water at 20–22°C, made opaque by adding white non-toxic paint, and allowed to swim for 60 s or until they found the location of a hidden circular platform (11 cm in diameter). The pool was virtually divided into 4 quadrants with 4 visible spatial cues different in color and shape. During the learning phase, mice were trained with 4 trials per day (starting from a different quadrant at each trial) for 4 days with 60 min inter-trial interval. For each trial, the time to reach the platform, the total distance traveled, and the mean speed were evaluated.

On the 5th day, the platform was placed in a new quadrant and pattern separation (reversal test) was evaluated as time spent in the new quadrant minus time spent in the quadrant where the platform was previously located. Finally, after reversal, mice were trained to learn the new platform position for 6 further consecutive days (D) before the probe test. On the 12th day, to assess memory retention, the platform was removed and the animals were allowed to swim for 60 s (probe test), and the latency to reach the platform area and total entries in the platform area were evaluated. The analysis of target search strategy was carried out according to current literature (**Curdt et al., 2022**). Briefly, spatial accurate strategy and non-spatial strategy were distinguished considering individual swim paths. The analysis was carried out at the end of the first training period (D4), in the post-reversal training period at D7, D9, and D11 and during the probe test.

## Three-chamber sociability

The three-chamber sociability (TCS) test was performed on 16 a.Ctrl (5 females, 11 males) and 17 a.218 (9 females, 8 males) animals at 2 months of age to evaluate sociability and interest in social novelty. The TCS apparatus is composed of one empty central chamber (20x30 x 30 cm) with dark walls and two lateral chambers (27x30 x 30 cm each) with a wire cup each, separated by removable doors. During the acclimation phase, mice were placed in the central chamber with both doors opened and allowed to freely explore for 10 min. Twenty-four hr after the first phase, animals were initially placed in the central area for 5 min with closed doors, then an unfamiliar mouse (stranger (S) 1) was placed inside the cup of one of the two lateral chambers while the other remained empty (object); the doors were opened and the testing animal could explore the whole apparatus for 10 min. After this session, each mouse was placed again in the central chamber with closed doors. A second unfamiliar mouse (S2) was allocated into the cup that was previously empty, doors were opened and the animal could explore freely for 10 min. In this phase, S1 represented the familiar mouse while S2 represented social novelty. In each session, S1 vs cup and S1 vs S2, the number of interactions with empty cups, the

latency to the first interaction as well as the time spent and number of entries in each chamber were considered.

## Marble burying

To evaluate compulsive/perseverative behavior and anxiety, a.Ctrl and a.218 mice were subjected to marble burying test (MBT) at 2 and 6 months of age. After a 30 min habituation phase, performed in cages (36x25 x 25 cm) with 5 cm of clean bedding material, 20 black marbles (1.5 cm diameter) were placed on top of the sawdust, equally distributed in 5 rows. Mice were allowed to explore freely for 15 min. During the latter phase, the number of buried marbles and the latency to the first burying event were evaluated. A marble was considered buried if covered for at least 2/3 of its surface (*Daini et al., 2021*).

## Stereotypies

Stereotyped behavior was evaluated by manually counting the number of grooming, rearing and digging events at 2 and 6 months of age in the first 5 min of the OF test and first 10 min of the MBT, respectively (*Daini et al., 2021*).

## ISH methods

### miR-218 detection in the CNS

For miR-218 detection, four C57Bl/6 J mice for each age (P2, P7, P60) were used. For histological processing, animals were anesthetized with isoflurane and brains from P2 and P7 mice were extracted from the skull and rapidly frozen using dry ice. All procedures were conducted in a Rnase free condition using Diethylpyrocarbonate (DEPC) pre-treated solutions. Intracardiac perfusion was performed only in P60 mice with 4% PFA in DEPC-Phosphate Buffered Saline (DEPC-PBS) (70 mL/7 min) preceded by an infusion of 50 mL of 0.9% NaCl saline containing heparin sodium (5000 U/L). The brains were postfixed in the same solution for 12 hr and rinsed in 20% sucrose-PBS-DEPC for 1 day. All brains were frozen using dry ice, and coronal 20-μm-thick sections were cut using a Cryostat, washed three times in cold PBS-DEPC and immediately processed for miRNA detection.

In situ hybridization (ISH) for fluorescent (FISH) and optical analyses was performed using miRCURY LNA microRNA ISH Optimization Kit and 5′ and 3′ digoxigenin (DIG) labeled LNA probes (Exiqon) as described by the manufacturer's instructions with minor modifications. Briefly, freshly sectioned brain tissue slices were post-fixed with 4% PFA for 10 min at room temperature followed by treatment with 3% $H_2O_2$. Then, immediately before use, Proteinase-K (PK) was diluted in PK buffer (5 μg/mL) and slides were incubated at 37 °C for 10 min. After three washes in DEPC-PBS at room temperature, slides were incubated with DIG-labeled-LNA-miR probe (Exiqon) overnight at the hybridization temperature (HybT) in microRNA ISH buffer: 60 °C for U6 and miR-124 probes, 56 °C for miR218 and scramble probes (day1). After stringent washes at HybT and blocking, slides were incubated with an anti-DIG-POD antibody (1:100; Roche) or anti-DIG-AP (1:200; Roche) for 1 hr at room temperature and developed, respectively, using Alexa Fluor 488-TSA (PerkinElmer) for 10 min or NBT/BCIP (Roche) for 4 hr at room temperature (day 2).

To determine miRNA expression in specific brain cell populations, FISH and IF protocols were coupled. Primary antibodies (Abs) against glial fibrillary acidic protein (GFAP, rabbit anti-GFAP (1:2000, Dako, Z0334, RRID:AB_10013382)) and neuronal nuclear antigen (NeuN, mouse anti-NeuN, 1:200, Millipore, MAB377, RRID:AB_2298772) and secondary Abs were added to slides during day 1 and day2 of ISH, respectively.

## Statistics

Statistical analyses were performed using GraphPad Prism or R. The Shapiro-Wilk test was used to test data for normal distribution. Normally-distributed data were analyzed with parametric tests (Student's t-test or ANOVA), otherwise non-parametric tests were used (Mann-Whitney or Kruskal-Wallis tests). Statistics for experiments with multiple observations (e.g., movies, tissue sections, or cells) from individual animals were derived using linear mixed models using the R packages lmerTest and emmeans to correct for multiple comparisons, as described in *Yu et al., 2022*. Individual details (tests and sample numbers) are described in the figure legends. All behavioral data, expressed as mean ± standard error

of the mean (SEM), were analyzed using Student's t test or one- or two-way ANOVA or Fisher's exact test. Differences were considered significant with $p < 0.05$. $0.05 < p < 0.1$ were also reported.

## Acknowledgements

We thank X Wang and G Leo for help with antagomiR injections; X Wang for help with viral preparation; B Lim for reagents; J Liu and B Ciarlo for help with calcium imaging analysis; N Amin and S Driscoll for help with KO mice and RNA-seq analysis; A Roberts, E Daini and M Bodria for help with behavioral analysis. Multi-photon laser scanning microscopy was performed at the Scripps Research Core Microscopy Facility, Nikon Center of Excellence, supported, in part, with funding from NIH HEI Grant 1 S10 OD026817-01. This work was supported by grants from the NIH (2R01NS012601, 1R21NS087342, 1R01NS121223, 1R01NS092705), TRDRP (22XT-0016, 21FT-0027), Whitehall Foundation (2018-12-55), MIUR (Department of excellence 2018–2022, E91I18001480001). NZ is supported by a fellowship from Autism Speaks (12923). AH was supported by a fellowship from the American Epilepsy Society (5–79096).

## Additional information

### Funding

| Funder | Grant reference number | Author |
| --- | --- | --- |
| National Institutes of Health | 1 S10 OD026817-01 | Giordano Lippi |
| National Institutes of Health | 2R01NS012601 | Darwin K Berg |
| National Institutes of Health | 1R21NS087342 | Darwin K Berg |
| National Institutes of Health | 1R01NS121223 | Giordano Lippi |
| National Institutes of Health | 1R01NS092705 | Christina Gross |
| Tobacco-Related Disease Research Program | 22XT-0016 | Darwin K Berg |
| Whitehall Foundation | 2018-12-55 | Giordano Lippi |
| Autism Speaks | 12923 | Norjin Zolboot |
| American Epilepsy Society | 5–79096 | Andrea Hartzell |
| Ministero dell'Istruzione, dell'Università e della Ricerca | E91I18001480001 | Michele Zoli |
| Tobacco-Related Disease Research Program | 21FT-0027 | Darwin K Berg |

The funders had no role in study design, data collection and interpretation, or the decision to submit the work for publication.

### Author contributions

Seth R Taylor, Conceptualization, Data curation, Formal analysis, Supervision, Investigation, Visualization, Methodology, Writing – original draft, Project administration, Writing – review and editing; Mariko Kobayashi, Conceptualization, Data curation, Formal analysis, Investigation, Visualization, Methodology, Writing – review and editing; Antonietta Vilella, Norjin Zolboot, Conceptualization, Data curation, Formal analysis, Investigation, Visualization, Methodology, Writing – original draft, Writing – review and editing; Durgesh Tiwari, Conceptualization, Data curation, Formal analysis, Investigation, Methodology, Writing – original draft, Writing – review and editing; Jessica X Du, Data curation, Formal analysis, Investigation; Kathryn R Spencer, Data curation, Investigation, Methodology; Andrea

Hartzell, Formal analysis, Supervision; Carol Girgiss, Yusuf T Abaci, Data curation; Yufeng Shao, Data curation, Formal analysis; Claudia De Sanctis, Resources; Gian Carlo Bellenchi, Resources, Writing – review and editing; Robert B Darnell, Supervision, Writing – review and editing; Christina Gross, Conceptualization, Supervision, Funding acquisition, Methodology, Project administration, Writing – review and editing; Michele Zoli, Conceptualization, Formal analysis, Supervision, Funding acquisition, Methodology, Writing – original draft, Writing – review and editing; Darwin K Berg, Conceptualization, Supervision, Funding acquisition; Giordano Lippi, Conceptualization, Supervision, Funding acquisition, Visualization, Writing – original draft, Project administration, Writing – review and editing

### Author ORCIDs
Robert B Darnell  http://orcid.org/0000-0002-5134-8088
Christina Gross  http://orcid.org/0000-0001-6057-2527
Giordano Lippi  http://orcid.org/0000-0003-3911-0525

### Ethics
All experimental procedures at UCSD, TSRI, CCHMH and SRI were performed as approved by the Institutional Animal Care and Use Committees and according to the National Institutes of Health Guidelines for the Care and Use of Laboratory Animals. Behavioral and in vivo experiments at UNIMORE were conducted in accordance with the European Community Council Directive (86/609/EEC) of November 24, 1986, and approved by the ethics committee (authorization number: 37/2018PR).

### Decision letter and Author response
Decision letter https://doi.org/10.7554/eLife.82729.sa1
Author response https://doi.org/10.7554/eLife.82729.sa2

## Additional files

### Supplementary files
• Supplementary file 1. Differential expression results from RNA-seq experiments. (1 A) Differential expression results (edgeR) from RNA-seq of dorsal hippocampi of P8 mice injected with a.218 vs a.Ctrl LNAs at P2. (1B) Differential expression results (edgeR) from RNA-seq of P15 *Mir218-1*$^{-/-}$; *Mir218-2*$^{+/-}$ vs wild-type mice. (1 C) Genes enriched in P8 VGluT2 +neurons from dorsal hippocampus, generated by differential expression comparison to genes enriched in P8 dorsal hippocampal VGAT +neurons (1D).

• MDAR checklist

### Data availability
RNA-seq data has been deposited to GEO (accession number GSE241245).

The following dataset was generated:

| Author(s) | Year | Dataset title | Dataset URL | Database and Identifier |
|-----------|------|---------------|-------------|-------------------------|
| Taylor S, Kobayashi M, Lippi G | 2023 | MicroRNA-218 instructs proper assembly of hippocampal networks | https://www.ncbi.nlm.nih.gov/geo/query/acc.cgi?acc=GSE241245 | NCBI Gene Expression Omnibus, GSE241245 |

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
