## [Editor Report]

This study describes important work documenting a role of microRNAs in regulating excitability in the developing mouse hippocampus. Strengths of the findings include the identification of several developmentally regulated microRNAs, specific manipulation in principle and inhibitory neurons, and multimodal analysis of developing circuits and gene expression. Solid methods are used to suggest a particular developmental role for interneurons and their regulation by microRNA-218.

---

## [Decision Letter]

**Decision letter after peer review:**

Thank you for submitting your article "MicroRNA-218 instructs proper assembly of hippocampal networks" for consideration by *eLife*. Your article has been reviewed by 2 peer reviewers, and the evaluation has been overseen by a Reviewing Editor and John Huguenard as the Senior Editor. The following individual involved in the review of your submission has agreed to reveal their identity: Gary Brennan (Reviewer #2).

Essential revisions:

1) There are major concerns related to the statistical assessment of the data being inappropriate in multiple instances.

2) Please address how the loss of miR-218 in the cellular phenotypes is causally involved in seizure susceptibility and behavior deficits.

3) Are seizures assessed at the chronic epileptic stage? the current data doesn't causally link changes in seizure susceptibility to miR-218 function in interneurons.

*Reviewer #1 (Recommendations for the authors):*

In this study, Taylor et al. describe the developmental role of miRNA-218 in shaping postnatal hippocampal circuits. They show developmental regulation of miR-218 expression and inhibiting miR-218 activity in early phases of post-natal developmental results in excessive network activity and susceptibility to epileptic seizures after kainate-induced insults. Conditional knock-out of miR-218 in interneurons indicates a possible cell-type specific role of miR-218 in shaping early-life hippocampal circuitry.

Describing a cell-type specific function of a particular microRNA in neural circuitry development during a defined time window would definitely represent a major advance in the field. However, in my opinion, the presented data falls significantly short in linking interneuron (IN) – and/or pyramidal neurons (PN)-expressed miR-218 to the observed structural, electrophysiological, or behavioural phenotypes. Moreover, the molecular mechanisms of how miR-218 regulates neural excitability, synapse structure, and behaviour remain almost completely obscure. Therefore, as outlined in more detail below, extensive additional data together with solidification of the already presented data, in particular regarding statistics, is required to warrant publication of this paper in *eLife*.

Specific comments:

1. The authors show that transient oligonucleotide-mediated miR-218 inhibition leads to deficits in interneurons and pyramidal neurons, but whether these deficits are cell-autonomous and which of these defects underlies the observed cognitive and/or behavioural phenotypes remains unknown. Therefore, it would be important to measure at least some of these parameters (i.e., spine density, intrinsic electrophysiological properties, stereotypic behaviour, memory) also in mice that lack miR-218 specifically in excitatory neurons (Vglut-Cre) or inhibitory (Vgat-Cre) interneurons.

2. Similarly, the current data doesn't causally link changes in seizure susceptibility to miR-218 function in interneurons. Simply showing that PTX treatment phenocopies miR-218 inhibition is not enough in this regard. Therefore, the authors have to repeat calcium imaging experiments (Figure 8) in slices from interneuron-specific (Vgat-Cre) miR-218 cko mice. Even more elegant would be a rescue experiment where miR-218 is re-expressed (e.g., using AAV-miR218-hairpin) specifically in interneurons of miR-218 deficient mice at P2 (and once the developmental window is closed, e.g. P21) followed by calcium imaging and the measurement of seizure susceptibility. Alternatively, the authors could inject a rAAV-CAMKII-218sponge into Vgat-Cre miR218 cko animals and see if susceptibility to epilepsy increases, e.g by performing EEG recordings. Such experiments would provide insight into a potential synergy of miR-218 function in IN versus PN.

3. Another major weak point of this manuscript is the lack of a target or mechanism for the described developmental role of miRNA-218. The authors performed elegant FACS-sorting and subsequent RNA-seq but do not exploit the data accordingly to mine for possible concrete targets and molecular mechanisms. We feel that at the very least the authors should show differential expression of a few of the most promising targets using multiplexed fluorescent in-situ hybridization assays with single-cell type resolution (INs vs PNs). In addition, a direct functional interaction of miR-218 with some of the predicted targets should be validated, e.g., using luciferase reporter assays. The ideal experiment would consist of a cell-type specific rescue using an AAV-RNAi approach against possible upregulated targets in the miR-218-2 cko line to try to rescue the altered network activity and/or seizure susceptibility.

4. The statistical assessment of the data is inappropriate in multiple instances. For example, the authors often take the number of technical replicates as n, which is not correct and overpowering the tests (cf. Yu et al., 2022, for correct use of statistical tests). In most cases, mixed-effect models would represent more appropriate tests, in particular with regard to the following figures:

– Figure 1A: It is not clear, what they tested in the sequencing or where this comes from (Lippi et al., 2016?). What were the conditions, statistics etc?

– Figure 1B: I would like to see some sort of statistical test, showing the significance of time as a fixed effect.

– Figure 1F-I: the number of n are videos, which kind of test has been chosen? Have they been corrected for blocking effects? See Yu et al., 2022 as a reference for correct statistics.

– Figure 2B-C: Overpowered with the use of the number of videos and the use of a simple Mann-Whitney test, assuming independence for all videos coming from one mouse. Block for mouse or average all videos coming from one mouse.

– Figure 4H and 6B: Number of n are sections coming from 3-4 animals. Same as before, take into account blocking effects.

– Figure 6D and 7B-D: We are aware that the standard for electrophysiological data is to perform statistics on the level of the number of cells, but statistically speaking it is not the correct approach. We would like to break this norm and ask for a corrected statistical test taking into account blocking effects at the mouse level.

– Figure 8B-F: see comments from Fig1F-I.

– Figure 9C: see comments from Fig4H. Also, a two-way ANOVA-type statistical test would be more appropriate in Figures 9A, C, D, and E.

---

## [Author Response]

Essential revisions:Reviewer #1 (Recommendations for the authors):Specific comments:1. The authors show that transient oligonucleotide-mediated miR-218 inhibition leads to deficits in interneurons and pyramidal neurons, but whether these deficits are cell-autonomous and which of these defects underlies the observed cognitive and/or behavioural phenotypes remains unknown. Therefore, it would be important to measure at least some of these parameters (i.e., spine density, intrinsic electrophysiological properties, stereotypic behaviour, memory) also in mice that lack miR-218 specifically in excitatory neurons (Vglut-Cre) or inhibitory (Vgat-Cre) interneurons.

We agree that identifying which cell autonomous effects cause the long-term miR-218 loss-of-function phenotypes is critical to understanding miR-218 roles instructing hippocampal circuit development. Because interneuron-specific miR-218 loss-of-function (VGat-Cre/miR-218^fl/fl^) phenocopies the changes in seizure susceptibility observed with a.218, we prioritized finding which phenotypes are linked to this cell type. We performed spine density analysis at P11, and marble burying behavioral analysis in the young adult in VGat-Cre/miR-218^fl/fl^ mice.

Interestingly, we found that these phenotypes are not altered when miR-218 is removed from interneurons only (Figure 9F, 10D), suggesting that miR-218 loss-of-function in pyramidal neurons, or maybe in both pyramidal neurons and interneurons simultaneously, is necessary for these phenotypes to emerge.

2. Similarly, the current data doesn't causally link changes in seizure susceptibility to miR-218 function in interneurons. Simply showing that PTX treatment phenocopies miR-218 inhibition is not enough in this regard. Therefore, the authors have to repeat calcium imaging experiments (Figure 8) in slices from interneuron-specific (Vgat-Cre) miR-218 cko mice. Even more elegant would be a rescue experiment where miR-218 is re-expressed (e.g., using AAV-miR218-hairpin) specifically in interneurons of miR-218 deficient mice at P2 (and once the developmental window is closed, e.g. P21) followed by calcium imaging and the measurement of seizure susceptibility. Alternatively, the authors could inject a rAAV-CAMKII-218sponge into Vgat-Cre miR218 cko animals and see if susceptibility to epilepsy increases, e.g by performing EEG recordings. Such experiments would provide insight into a potential synergy of miR-218 function in IN versus PN.

As suggested by this reviewer, we performed calcium imaging experiments from VGat-Cre/miR218^fl/fl^ mice. At P5, miR-218 loss-of-function in interneurons phenocopied the decrease in synchronous activity observed with a.218 (Figure 10A-C). This is consistent with a body of literature suggesting that GABA released from interneurons is the major source of synchronous activity at this developmental stage. We conclude that miR-218 is a critical regulator of interneuron function in the developing hippocampus.

Surprisingly, we observed a similar phenotype at P8 (Figure 10A-C). This is the opposite phenotype of what we observed with a.218 (Figure 1F-I, 8E-F), suggesting that blockade of miR-218 function in pyramidal neurons and interneurons simultaneously results in a complex compound phenotype that is not due to the function of miR-218 in interneurons alone.

3. Another major weak point of this manuscript is the lack of a target or mechanism for the described developmental role of miRNA-218. The authors performed elegant FACS-sorting and subsequent RNA-seq but do not exploit the data accordingly to mine for possible concrete targets and molecular mechanisms. We feel that at the very least the authors should show differential expression of a few of the most promising targets using multiplexed fluorescent in-situ hybridization assays with single-cell type resolution (INs vs PNs). In addition, a direct functional interaction of miR-218 with some of the predicted targets should be validated, e.g., using luciferase reporter assays. The ideal experiment would consist of a cell-type specific rescue using an AAV-RNAi approach against possible upregulated targets in the miR-218-2 cko line to try to rescue the altered network activity and/or seizure susceptibility.

To identify miR-218 targets that are responsible for the a.218 phenotypes, we first performed qPCR analysis on the candidates that were consistently upregulated in RNAseq experiments. Two genes, Kif21b and Syt13, were confirmed as significantly upregulated (Figure 10E). As we did before (Lippi et al. 2016), we tested if their selective and simultaneous de-repression with target site blockers (TSB) phenocopied aspects of the a.218 phenotype. TSB-treated mice showed increased synchronous and total events at P8 (Figure 10F-G), suggesting that derepression of these two genes is an important driver of the altered spontaneous activity induced by a.218.

4. The statistical assessment of the data is inappropriate in multiple instances. For example, the authors often take the number of technical replicates as n, which is not correct and overpowering the tests (cf. Yu et al., 2022, for correct use of statistical tests). In most cases, mixed-effect models would represent more appropriate tests, in particular with regard to the following figures:

We appreciate the reviewer directing us Yu et al., 2022. Accordingly, we have revised the statistical analyses to account for the animal from which an observation is made. In many cases, we have now analyzed the data using linear mixed-effect models with animal-specific random effects, as recommended by Yu et al., 2022. The majority of our findings remained statistically significant with the revised statistical tests. In the few cases that this wasn’t true, we changed the text accordingly.

– Figure 1A: It is not clear, what they tested in the sequencing or where this comes from (Lippi et al., 2016?). What were the conditions, statistics etc?

We apologize for the lack of clarity. The RNA-Seq data are indeed from Lippi et al. 2016; we added a note in the figure legend. These are from 6 wild-type animals at P12. The numbers for each miRNA represent the percentage of the total reads and the standard deviation across the RNA-Seq replicates.

– Figure 1B: I would like to see some sort of statistical test, showing the significance of time as a fixed effect.

We added statistics to Figure 1B.

– Figure 1F-I: the number of n are videos, which kind of test has been chosen? Have they been corrected for blocking effects? See Yu et al., 2022 as a reference for correct statistics.

We have now analyzed these data with a linear mixed-effect model with animal-specific random effects. The updated statistics are described in the figure legend.

– Figure 2B-C: Overpowered with the use of the number of videos and the use of a simple Mann-Whitney test, assuming independence for all videos coming from one mouse. Block for mouse or average all videos coming from one mouse.

We have now analyzed these data with a linear mixed-effect model with animal-specific random effects. The updated statistics are described in the figure legend.

– Figure 4H and 6B: Number of n are sections coming from 3-4 animals. Same as before, take into account blocking effects.

We have now analyzed these data with a linear mixed-effect model with animal-specific random effects. The updated statistics are described in the figure legend.

– Figure 6D and 7B-D: We are aware that the standard for electrophysiological data is to perform statistics on the level of the number of cells, but statistically speaking it is not the correct approach. We would like to break this norm and ask for a corrected statistical test taking into account blocking effects at the mouse level.

We have now analyzed these data with a linear mixed-effect model with animal-specific random effects. The updated statistics are described in the figure legend.

– Figure 8B-F: see comments from Fig1F-I.

We have now analyzed these data with a linear mixed-effect model with animal-specific random effects. The updated statistics are described in the figure legend.

– Figure 9C: see comments from Fig4H. Also, a two-way ANOVA-type statistical test would be more appropriate in Figures 9A, C, D, and E.

As recommended, we have performed new statistical analysis of Figure 9A and C. We respectfully disagree with the reviewer that a two-way ANOVA would be more appropriate for Figure 9D and E because the test is used to know how two independent variables, in combination, affect a dependent variable. In this case the knockout of miR-218 either in pyramidal neurons or in interneurons is not in combination. These are not animals coming from the same litters and experiments were not done the same day. The two data sets were grouped in a single bar graph for simplicity of visualization, but we realized it was misleading and confusing; hence, we split them in two graphs.